# Structure induced laminar vortices control anomalous dispersion in porous media

Ankur Deep Bordoloi [1✉], David Scheidweiler[1], Marco Dentz [2], Mohammed Bouabdellaoui[3], Marco Abbarchi [3] & Pietro de Anna [1✉]

Natural porous systems, such as soil, membranes, and biological tissues comprise disordered structures characterized by dead-end pores connected to a network of percolating channels. The release and dispersion of particles, solutes, and microorganisms from such features is key for a broad range of environmental and medical applications including soil remediation, filtration and drug delivery. Yet, owing to the stagnant and opaque nature of these disordered systems, the role of microscopic structure and flow on the dispersion of particles and solutes remains poorly understood. Here, we use a microfluidic model system that features a pore structure characterized by distributed dead-ends to determine how particles are transported, retained and dispersed. We observe strong tailing of arrival time distributions at the outlet of the medium characterized by power-law decay with an exponent of 2/3. Using numerical simulations and an analytical model, we link this behavior to particles initially located within dead-end pores, and explain the tailing exponent with a hopping across and rolling along the streamlines of vortices within dead-end pores. We quantify such anomalous dispersal by a stochastic model that predicts the full evolution of arrival times. Our results demonstrate how microscopic flow structures can impact macroscopic particle transport.

[1] Institute of Earth Sciences, University of Lausanne, Lausanne 1015, Switzerland. [2] Spanish National Research Council (IDAEA-CSIC), Barcelona 08034, Spain. [3] Aix Marseille Univ, Université de Toulon, CNRS, IM2NP, 13397 Marseille, France. ✉email: ankurdeep.bordoloi@unil.ch; pietro.deanna@unil.ch

Most geological and biological systems are characterized by a porous structure where fluids can move through a network of small confined spaces, also known as pores[1]. The interest of particle transport through porous media is manifold, and it spans across various natural systems and technological applications[2]. On the one hand, the broad range of variability in pore-size induces velocity heterogeneity leading to anomalous transport[3–5], mixing[6,7], filtration[8,9], and microbial dispersal[10,11]. On the other hand, morphological diversity introduced by disordered pore structures[12–14] induces a rich flow organization with spatial and temporal complexities that can play critical roles in groundwater contamination and remediation[15], enhanced hydrocarbon recovery[16], transport through river sediments that create a closely packed pore network[17], water filtration systems[18] and extra-cellular transport in brain tissue[19]. The complex morphology of a permeable system typically exhibits dead-end pores, the portion of the system that cannot host a net fluid transfer resulting in stagnant flow[20–22]. Such structures characterize soil[22], gut tissue[23], and polymeric filters[24]. Because of the stagnant nature of the fluid, it is common to assume that the transported quantities remain trapped within these dead-end-pores for long times until molecular diffusion allows them to escape[25,26]. Hence, a complete quantitative understanding of the role played by the local flow structures on the anomalous transport associated to dead-end-pores has remained overlooked to date.

To unravel the link between microscopic motion within dead-end-pores and porous media transport, we consider a medium composed of a network of transmitting-pores (TP) of similar size and randomly distributed dead-end-pores (DEP). As an archetype of such structural heterogeneity, we exploit spinodal-like morphology, that emerge from first-order phase transitions[27] in binary systems featuring a miscibility gap and undergoing phase separation, as described by Cahn-Hilliard dynamics originally developed for metallic alloys[28–30]. Such morphology of connected structures are the common ground for a plethora of natural and artificial systems undergoing spontaneous pattern formation, including volcanic rocks[31,32], semiconductors[33–35], metallic thin films[36], metal hydrides[37–41], liquid-gas[42], polymers[43–46], proteins[47] and tumor cells[48,49].

We build a model porous structure of such complexity by exploiting the method of solid-state dewetting[35]: it is based on annealing of mono-crystalline semiconductor thin films that results in surface morphology exhibiting spinodal topology and a disordered hyperuniform character, see Fig. 1b. The dewetted islands of this geometry constitute the impermeable matrix of a relatively homogeneous porous medium composed of TP (about 91% of the total volume) and DEP (about 9% of the total volume) with structural disorder that is statistically isotropic, similar to crystals or jammed disordered spheres packing[50]. Despite its overall homogeneity in terms of TP-size $\lambda$ (i.e., the distribution of $\lambda$-values is very narrow; see Fig. 1c), we show that this porous structure has widely distributed heterogeneous DEP-size and displays complex flow and transport properties that directly link to the underlying porous structure. Using microfluidic experiments and rigorous simulations we find a quantitative link between the anomalous macroscopic transport and the complex microscopic flow structures in the dead-end pores.

## Results

**Structure characterisation.** We use soft lithography[51] to print the spinodal porous structure (small sections shown in Fig. 1a, c, d) into a silicon wafer that acts as a mold for a PDMS microfluidic chip (see Supplementary Information I). The irregular grain features of the structure represent the solid matrix of the porous

micro-channel, characterized by a porosity $\phi = 0.39$. Between the inlet and outlet zones, the microfluidic channel has length $L = 50$ mm, width $w = 7$ mm and thickness $a = 0.083$ mm (corresponding to 1850 by 260 average pore sizes $\lambda_m = 0.027$ mm), schematically shown in Fig. 2b. The thickness $a$ is designed to be of the same order of magnitude as the average pore size, $\lambda_m = 0.027$ mm, yielding velocity profiles similar to a channel flow in the TP, as verified by PIV (see Supplementary Fig. 1). The local pore size $\lambda$ is computed using the method of maximum inscribed circle (MIC, see Methods $a$) across the whole domain. The medium is quite homogeneous in pore-size, such that the statistical distribution of $\lambda$ is narrow and has a strong peak close to the mean pore-size $\lambda_m$ (Fig. 1b), typically found in many geological structures[1]. We characterize the porous structure in terms of the segregation index $\zeta$ (see Methods $a$ and Fig. 1c, d) to distinguish between TP and DEP, respectively shown as green and magenta regions in Fig. 1d. Since $\zeta$ refers to the number of individual grains surrounding each pore, it captures the dual feature of the matrix (TP: $\zeta > 1$ and DEP: $\zeta = 1$). The DEPs are elongated pores caving inside a singular grain in contrast to the TPs that are surrounded by multiple grains. We quantify the aspect ratio $\Lambda$ of each DEP as the ratio between their area $\mathcal{A}$ and $\lambda_m^2$: it results that $\Lambda = \mathcal{A}/\lambda_m^2$ spans within the range $1-30$ and follows a power-law distribution with an exponential cut-off (i.e., a Gamma-distribution) (Fig. 1e).

**Macroscopic transport experiment.** We investigate the impact of this dual feature of the medium (TP and DEP) on the dispersion of a neutrally buoyant suspension of colloidal particles initially distributed within the porous matrix. To this end, we first produce a nearly-homogeneous distribution of suspended colloids across the medium by continuously injecting a density matched suspension (Solution A: polystyrene micro-spheres of diameter $0.5 \mu m$ and density $\rho = 1.05$ g/mL Thermofischer Fluoromax B500 in 1:1 milliQ water and $D_2O$ mixture) through the microfluidic channel at constant flow rate $Q = 0.2 \mu L/min$ for 24 h (~165 pore volumes). This allows a fraction of particles to escape the main stream flow and occupy the dead-end pores, mostly by diffusion. We quantify the fractional particle number density in DEP ($\alpha = n_{DEP}/n_{total}$) by analyzing the fluorescence signal in the microscopic image of the medium after 24 h and identifying particles trapped inside the DEPs (see Supplementary Information V and Fig. 9). The measured $\alpha = 0.22$ is approximately twice the volume fraction of total DEPs inside the system. This difference is attributed to the fact that, towards the end of the 24 h, some particles in the middle of the injection syringe have settled under gravity causing dilution to the initial concentration of the particle.

Next, we withdraw a clean solution with the same density (Solution B: 1:1 milliQ water and $D_2O$ mixture) but without the colloidal suspension through the inlet via a cleaning circuit (see 2a B-2-1-4): the medium outlet stays closed while we withdraw liquid from a hole next to the inlet, as discussed in[11]. This generates a particle-free sharp front at the inlet without perturbing the particles within the porous matrix. The solution $B$ is, then, withdrawn at a constant flow rate $Q = 0.5 \mu L/min$ through the breakthrough circuit (see 2a B-2-3-4) eluting one pore volume (i.e., the volume of the entire porous channel) every $t_{PV} = 21.8$ min, for about $60 t_{PV}$. The relative importance of advection to diffusion process is quantified by the corresponding Péclet number, $Pe = \lambda_m u_m/D$ based on the mean fluid velocity ($u_m = 0.038$ mm/s: the flow regime is laminar since $Re = u_m \lambda_m/\nu \sim 0.001$, where $\nu = 1$ mm$^2$/s) and the suspension diffusion coefficient, $D = 1.4 \times 10^{-7}$ mm$^2$/s, such that $Pe = 7329$. We independently measure the diffusivity of $0.5 \mu m$ polystyrene colloids in the studied medium and find $D$ to be smaller than the theoretical prediction for suspension in the bulk

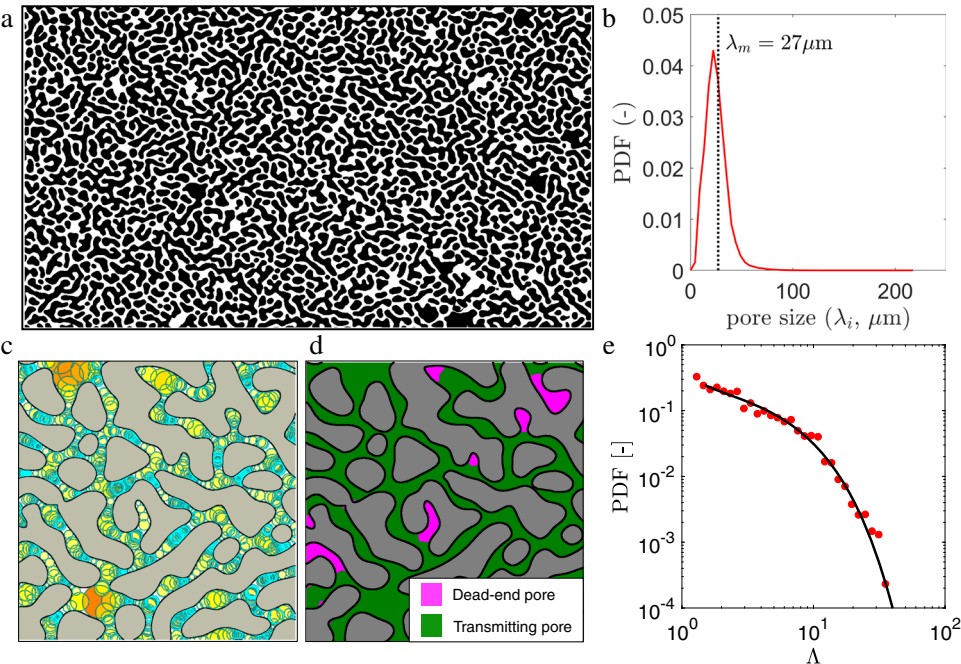

**Fig. 1 Characterization of the model porous structure reveals dual feature of the complex medium. a** Binary image of the disordered hyperuniform porous structure: it exhibits a complex pore network (white) interspersed among disordered grains (black) within the system. **b** The narrow Probability Density Function (PDF) of the pore size, exhibiting a strong peak about the average value $\lambda_m = 27 \, \mu m$. **c** A portion of the pore size map ($\lambda$, [$\mu m$]) highlighting the inscribed circles (cyan) along the porous network that estimate the local pore-size (see Methods a). **d** The dual feature of the medium characterized by the transmitting-pores (TP: green) surrounded by multiple grains and the dead-end pores (DEP: magenta) surrounded by a single grain. **e** Red dots represent the measured PDF of width to depth aspect ratio for DEPs, defined as the ratios between each DEP area ($\mathcal{A}$) and the mean pore-space area ($\lambda_m^2$); the solid black line is its best fit with a Gamma distribution $f_\Lambda$.

from the Stokes-Einstein equation due to the confined structure of the system (see Supplementary Information II).

Time-lapse composite imaging coupled with fluorescence microscopy (see Methods b) allows us to count the number of effluent colloids near the outlet and measure the breakthrough curve (BTC) over three orders of magnitude. The latter, displayed in Fig. 2c, shows two distinct transport regimes. The first regime, limited to the elution of about two pore-volumes, is well captured by the classical advection-dispersion framework[2] (see Methods d). The dispersion coefficient has been fitted and results to be $D^* = 0.03 \, mm^2/s$ (green dashed line in 2c) using a least-squares method (see Methods d). At later times ($t/t_{PV} > 2$), a second regime emerges, where the BTC shows a power-law-like heavy tail, reminiscent of the so-called anomalous transport behavior (e.g.,[2]). The power-law exponent is steeper than what would be expected of a solely diffusion based transport, wherein the BTC should obey a 1/2 power-law decay.

We also monitor the spatial distribution of suspended colloids within the porous system (see Fig. 2e) by periodically collecting two composite images of the entire microfluidic channel (composing $38 \times 9$ individual pictures) 7 times (0, 1, 6, 10, 16, 22 and 44 h after injection), each temporally separated by $\Delta t = 6 \, min$. By comparing each couple of consecutive images, we distinguish the suspended particles from those retained by the host solid structure (see Methods e). Figure 2d shows that all suspended particles (in red) in TP have already disappeared from the field of view after 6 h (~18 pore volumes). On the other hand, those in the DEP stayed retained, suggesting that the heavy tail in the BTC is contributed by the particles in DEP. Figure 2e shows that the spatial profiles of the deposited particles (top) at three different times ($t = 0, 1 \, h, 6 \, h$) remain constant (thus, no deposition takes place during the flow experiment), while those of suspended particles (bottom) decay with time homogeneously across the porous channel.

**Pore-scale simulations.** Because the fluid velocities within the DEPs are significantly smaller than the ones within the TPs, a detailed understanding of such velocity field requires a multi-scale description. Although Eulerian velocity measurement techniques such as Particle Image Velocimetry (PIV) has been used to quantify the fluid velocity inside cavities[52], unfortunately it is challenging to simultaneously resolve all the scales of such flow-fields in high-resolution using such methods (see Supplementary Fig. 1). Herein, we use COMSOL Multiphysics to numerically solve the two-dimensional steady state incompreensible Stokes flow equations in a subsection of the microfluidics geometry (see Methods f and Supplementary Fig. 4). The domain of this numerical computation is approximately one-fifth in length (11 mm) and the same in width (7 mm), as that in the experiment. The computed velocity magnitude is resolved upto several orders of magnitude and it is shown in Fig. 3a in logarithmic scale (increasing velocity from light to dark). Further, we simulate the Lagrangian trajectories of $N = 10^5$ particles initially distributed across the entire medium with the same fractional number density in DEPs ($\alpha = 0.21$) as that in the experiment. The particles are transported by a combination of the computed velocity field and molecular diffusion (see Methods f), via the Langevin equation:

$$\vec{x}(t + \Delta t) = \vec{x}(t) + \vec{v}[\vec{x}(t)]\Delta t + \sqrt{2D\Delta t}\xi(t), \qquad (1)$$

where $\vec{x}(t)$ is the particle position at time $t$, $\vec{v}(\vec{x})$ is the local flow velocity (obtained by interpolating the computed velocity values to the particle location[3]), and $\xi(t)$ is a Gaussian random number with zero mean and unit variance. We consider three Péclet numbers ($Pe = 68, 680, 6800$) in these simulations by changing the diffusion coefficient by three orders of magnitude. We compute the BTC of the simulated particles resolved to four orders of magnitude (see Fig. 3b) that shows the same

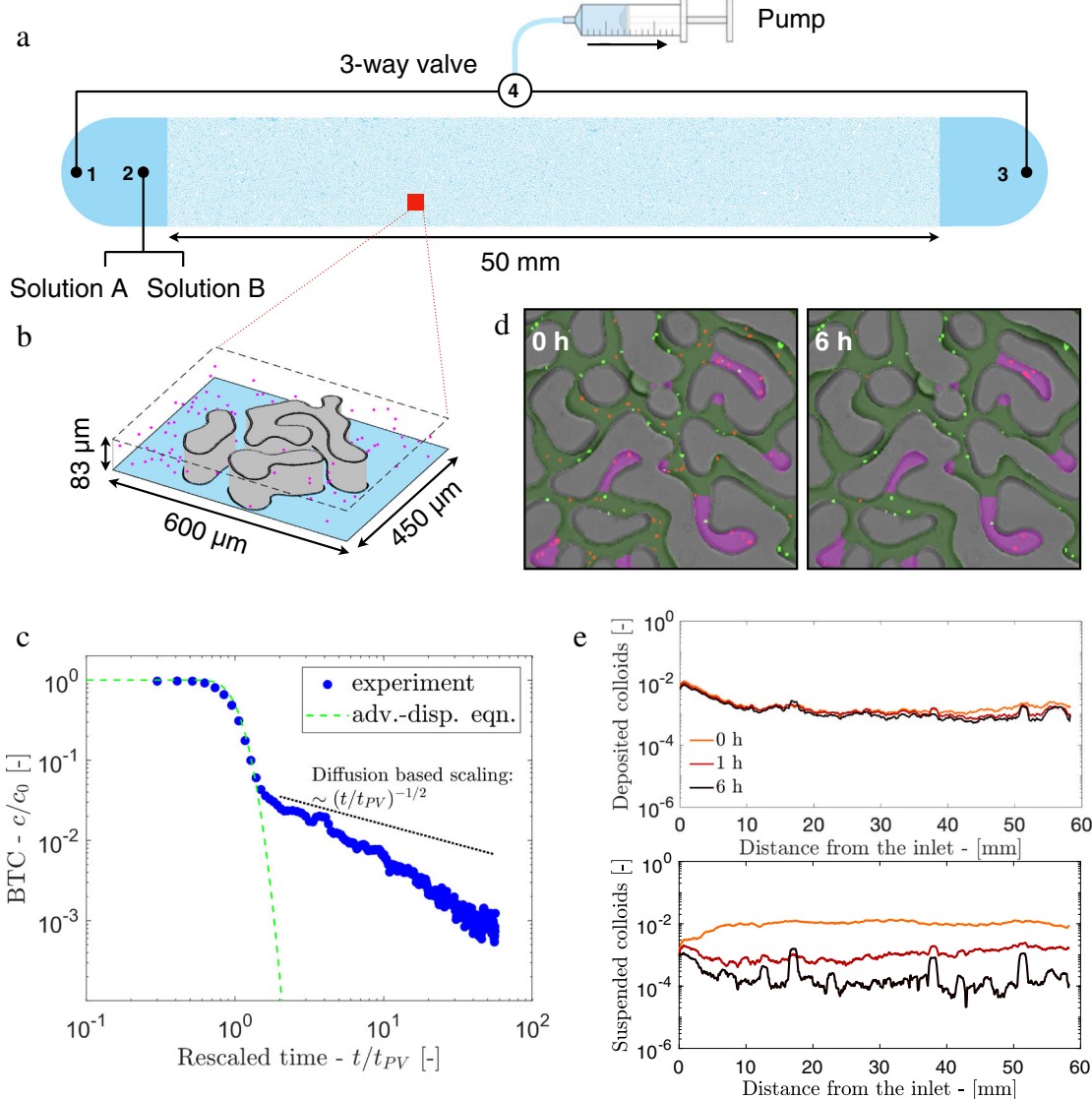

**Fig. 2 Dual geometric feature of the medium leads to two distinct regimes in the breakthrough curve of colloidal particles. a** Schematic of the experimental setup and (**b**) three-dimensional representation of the colloidal suspension within the porous structure. **c** Experimental breakthrough curve (BTC, blue dots) computed as the $c(t)/c_0$, where $c_0$ is the injected colloidal numerical density, and $c(t)$ is the measured density eluted at time $t$. The dashed line represents the analytical solution of advection-dispersion equation (Eq. 26 in Supplementary Information III). The dotted line represents the prediction from diffusion based scaling (without advection). **d** Two snapshots of suspended (red) and deposited (green) colloids just before and 6 h after injection, respectively; TPs are represented as green areas while DEPs as magenta. **e** Profile of deposited (above) and suspended particles (below) along the channel length acquired at three different times (0, 1, and 6 h) after the start of the experiment.

two distinct regimes observed in our experiment. The late-time tail of the BTC depends on the global Péclet number ($Pe$), such that a decreasing $Pe$ results in a shorter tail. We investigate further into the two regimes for $Pe = 680$ (shown in green and magenta shades in Fig. 3b) by finding the initial positions of the particles with the same colors in the inset of Fig. 3a. It is clear that the late time scaling in the BTC is due to the particles initially trapped within the DEPs, whereas the particles not in the DEPs are washed-out from the medium before the transition time, about $t/t_{PV} = 2$, as observed experimentally. In addition, we consider the possibility of a particle re-entering into a DEP, by counting such instances (see Fig. 3c). The re-entry event is significantly small (<0.1%) due to: a) only a small fraction of particles get close to the DEP entrance and b) the total volume fraction of DEP covers less than 10% of the total porous matrix.

**Structure induced vortices and pore-scale transport.** Due to the low fluid velocity inside a DEP, it is common to assume that the DEPs delay the macroscopic transport by diffusive trapping and release. As a consequence, porous systems comprising of DEPs are generally modeled as dual media with separate advection-dominated and diffusion-dominated regions (e.g.,[53]). These models predict that particles move fast through the advection-dominated region, or they spend long times trapped in a diffusion-dominated region, and their transition across the two regions is characterized by a given transfer rate. Contrary to this simplified view about such systems, we show the existence of complex vortex flow structures characterized by flow recirculation within the DEPs that, coupled with molecular diffusion, control the macroscopic BTC. To qualitatively examine the nature of these flow structures, Fig. 3d displays a time-stacking image of transported colloidal particles for $Pe \sim 10^5$ inside a representative

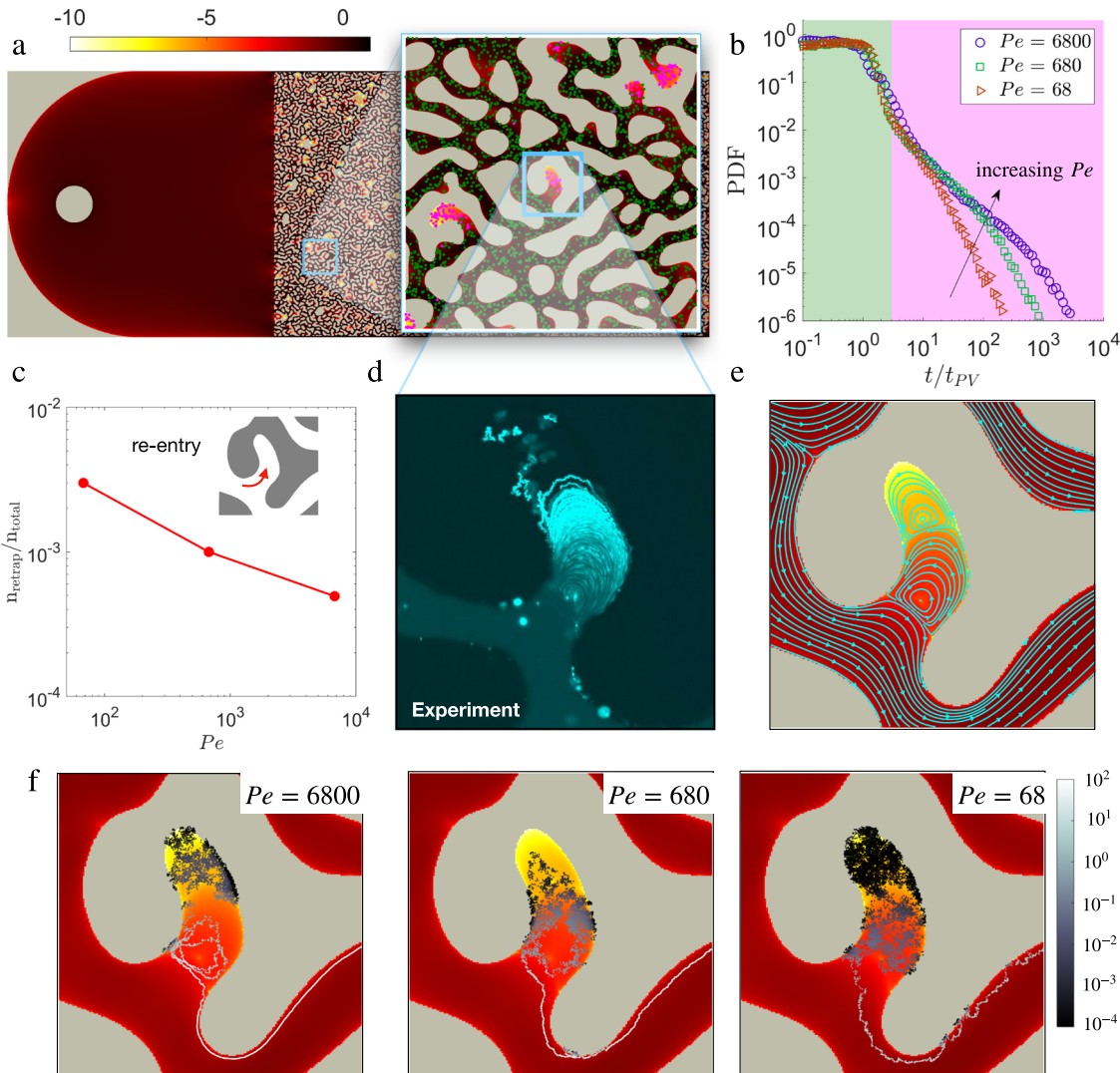

**Fig. 3 Computation of velocity field and the role of structure induced vortices on particle dispersal. a** Modulus of the Stokes flow solutions (mm/s in log-scale) in a subsection (1/5$^{th}$ in length) of the porous medium used in the experiment is superposed to particles that initially occupy the TP (green) and DEP (magenta) (enlarged view in the inset). **b** Probability density function (PDF) of particle escape time (equivalent to the BTC) versus normalized time ($t/t_{PV}$) obtained from particle tracking in the simulated velocity field with $\alpha = 0.22$ for three Péclet numbers ($Pe = 68, 680, 6800$). The magenta and green shades distinguish the regions of the BTC for $Pe = 680$ contributed by the particles shown in corresponding colors in the inset of (**a**). The long tail of the PDF is contributed by particles originating in the DEPs. **c** Fraction of particle number density re-entering a dead-end pore for the three Péclet numbers. **d** Qualitative identification of vortex structure inside a DEP captured by time-stacking experimental images taken at $Pe \sim 10^5$. **e** Close view of the vortex structures inside a DEP from the simulated velocity field. **f** An individual trajectory of a particle originated at the magenta dot and leaving the DEP for $Pe = 6800, 680,$ and $68$. The trajectories are color-coded with a local Péclet number $Pe^* = \lambda_m v_p/D_m$ in log-scale.

DEP. At the entrance of the DEP, where the velocity is smaller than in the adjacent TP, a laminar vortex is identified by a collection of closed trajectories. Far inside the DEP, the individual tracks of random walkers are visible as short and tortuous segments suggesting the dominance of molecular diffusion.

We use the velocity fields from the simulations explore such structure of flow inside a DEP. It is similar to the one of the classical cavity flows, characterized by a cascade of vortices with decaying velocity magnitude that spans a very broad range of scales[54]. A close-up view of the velocity field magnitude inside a DEP (see Fig. 3e) superposed with a few characteristic streamlines (in cyan) shows the underlying flow structure organized in a cascade of laminar vortices. In this scenario, a particle can escape a vortex of closed streamlines only by hopping across them via molecular diffusion. This mechanism is demonstrated in Fig. 3f by plotting the trajectory of a particle initiated deep inside a DEP

for three Péclet numbers ($Pe = 6800, 680, 68$). The trajectory is color-coded with a local Péclet number (increasing in values from dark to light) defined as $Pe^* = \lambda_m v_p/D$, where $v_p$ is the local velocity of the tracked particle. The effect of the vortex roll on the trajectory in this specific DEP is most prominent for $Pe = 6800$. Initially being in the low velocity zone ($v_p \sim 10^{-7}$ mm/s), the particle is transported randomly across streamlines by molecular diffusion ($Pe^* \sim 10^{-4}$), and as it approaches the outer vortex, it hops across streamlines but also rolls along the vortex ($Pe^* \sim 1$), before being advected away by the TP flow ($Pe^* \sim 10^2$). When the Péclet number is small ($Pe = 68$), the diffusion of the particle across streamlines becomes more pronounced with lesser advective rolling along the vortex. These comparisons suggest that the increasingly heavier anomalous tail in the BTC for the larger $Pe$ (see Fig. 3b) is due to a combination of rolling and hopping across streamlines in the vortex near the DEP entrance.

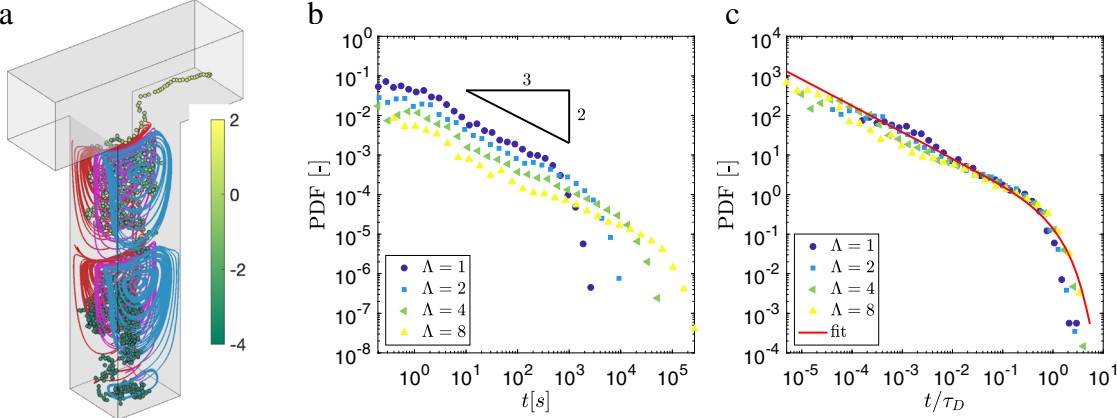

**Fig. 4 Conceptual model for DEP flow capturing the local particle escape time. a** A single numerically simulated trajectory (color-coded with local Péclet number $Pe^* = \lambda_m v_p / D_m$ in log-scale) originating at the bottom of a 3D rectangular cavity representing a DEP (aspect ratio, $\Lambda = 4$) connected to a square channel representing a TP. A series of streamlines highlights the vortex flow structure inside the cavity. **b** Particles escape time probability density function (PDF, equivalent to their BTC) of a single cavity for $\Lambda = 1, 2, 4, 8$. **c** The same as (**b**) with time rescaled by diffusion time-scale $\tau_D = (\lambda_m \Lambda)^2 / D$.

**Bridging from pore-scale features to macroscopic transport**. To understand how this mechanism of rolling along and hopping across closed streamlines controls the late time behavior of the BTC, we first model transport in a single DEP generalizing its geometry to a cavity connected to a free channel. The widths ($w$) of cavity and free channel are set to be equal to $\lambda_m$, and the cavity-size ($l_d$) is varied based on four aspect ratios $\Lambda = l_d / w = 1, 2, 4, 8$. We perform this simulation for both two- and three-dimensional geometries, adding a uniform thickness of $\lambda_m$ to the latter. Using the same numerical scheme adopted to solve the transport in Fig. 3, we compute $10^4$ particle-trajectories initiated homogeneously inside each cavity volume. Figure 4a shows the result for a 3-dimensional representative case with $\Lambda = 4$: the flow structure is highlighted by multiple streamlines (initiated at three $y$-plane locations shown in different colors) associated with the vortices of decaying intensity along the depth[54]. The trajectory of a particle initiated at the cavity bottom is shown with the same color-scheme as in Fig. 3d. The trajectories for both 3D and 2D cavities (see also Supplementary Fig. 6) show behavior similar to the ones in the DEP of the porous medium: the particle initially diffuses isotropically ($Pe^* \ll 1$) until it reaches the upper part of the cavity, where its motion is a result of the competition between advection along the vortex streamlines and diffusion that promotes streamline exchange ($Pe^* \sim 1$). Finally, the particle exits the cavity and follows the channel (TP) flow with $Pe^* \gg 1$.

Next, we compute the BTC for the four tracking simulations, $\Lambda = 1, 2, 4, 8$, as the PDF of particle arrival time to the channel outlet. Figure 4b shows that, for all cases the distribution decays as the power law $t^{-2/3}$ with an exponential cut-off: a Gamma distribution. This power-law exponent is different from the scaling expected for diffusion alone, $t^{-1/2}$[55]. The cut-off time for each $\Lambda$ is given by the characteristic diffusion time over the DEP depth, such that $\tau_D = (\lambda_m \Lambda)^2 / D$. Hence, when time is rescaled by $\tau_D$, the BTCs for all aspect ratios $\Lambda$ collapse into the master-curve

$$g(t) = \frac{(t/\tau_D)^{-2/3} \exp(-t/\tau_D)}{\Gamma(1/3)}. \quad (2)$$

The BTCs obtained for 2D cavities also exhibit an identical collapse (see Supplementary Fig. 6). This suggests that the thickness of the cavity does not affect the transport of particles from a dead-end pore. Additionally, we examine the effect of the DEP shape by considering a tortuous cavity. We find that

the tortuousity of a DEP does not affect the BTC (see Supplementary Fig. 7). The shape-independent characteristic is attributed to the fact that the power-law scaling in the BTC is primarily controlled by the fastest vortex near the entrance.

The characteristic scaling of $t^{-2/3}$ is controlled by the trapping of particles in closed streamlines. The combined action of advection-controlled transport in the shear flow along the solid boundaries within a vortex, and diffusion at the open boundaries of the vortex itself leads to a scaling of residence times as $\psi(t) \sim t^{-1-\gamma}$ with $\gamma = 2/3$, as discussed in a different context by[56]. This residence time distribution corresponds to a trapping time distribution scaling as $t^{-\gamma}$. A detailed derivation of this scaling exponent is provided in the Supplementary Information III.

Based on the above scaling, we formulate a 1D statistical model (illustrated in Fig. 5a) by considering the porous system as a combination of several DEPs with a distribution of aspect ratio $\Lambda$. The details of this model are described in the Supplementary Information IV. The distribution of $\Lambda$ is obtained from the characterization of the medium as in Fig. 1e, results in a similar distribution in the time scale $\tau_D$. The global BTC can be constructed as the weighted average between the BTC of particles originated within TPs (solution of advection-disperion equation: $F_{TP}$) and the BTC of particles originated within DEPs ($F_{DEP}$). The latter is obtained from the above Gamma distribution (Eq. (2)) weighted by the probability density function $f_D(t)$ of the corresponding characteristic diffusion time $\tau_D$,

$$F(t) = \underbrace{(1-\alpha)\frac{1}{L}\int_0^L dx f_0(t,x)}_{F_{TP}} + \underbrace{\alpha \int_0^\infty d\tau g(t/\tau)\tau^{-1} f_D(\tau)}_{F_{DEP}}. \quad (3)$$

Here, $\alpha$ is the fraction of observed colloids initially located in DEP and $1 - \alpha$ the fraction of those initially located in TPs. The function $f_D(t)$ can be expressed in terms of the measured PDF $f_\Lambda(a)$ of aspect ratios $\Lambda$ as $f_D(t) = f_\Lambda\left(\sqrt{tD/\lambda^2}\right)\sqrt{D/(\lambda^2 t)}/2$, and $f_0$ is the solution of the advection-dispersion equation representing particle originating in TPs (see Methods and Supplementary Information III). The model, which is completely based on the medium geometry, initial volume fraction of colloids in DEP, and the flow characteristics, show excellent agreement with the experiment (Fig. 5b) and simulation results (see Supplementary Fig. 5).

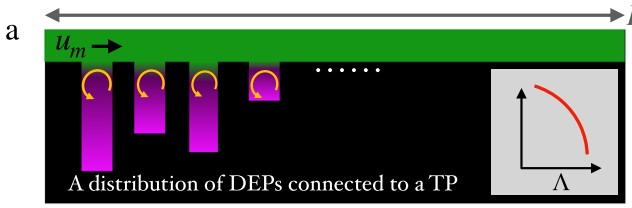

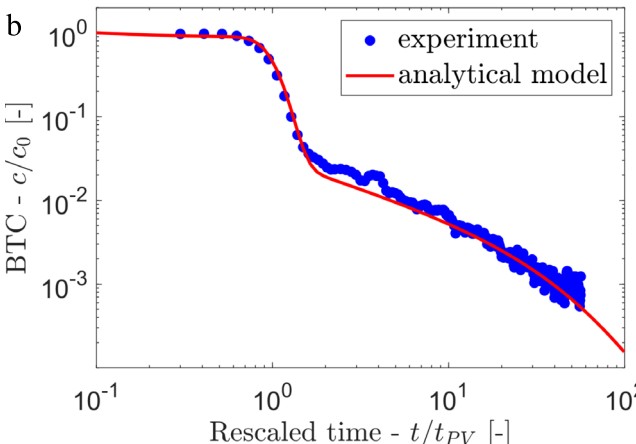

**Fig. 5 Upscale from pore-scale features to macroscopic transport.**
**a** Schematic representation of a one-dimensional analytical model described by a straight TP connected to a collection of DEPs via Eq. (3); the distribution of DEP-size is parameterized by the PDF of aspect ratio ($\Lambda$), and their number density by the parameter $\alpha$. **b** Comparison of the analytical prediction (solid line) from Eq. (3) of the BTC with the experimental data (dots).

## Discussion

In addition to molecular diffusion of the transported particles and the average flow (i.e., the Péclet number), the observed anomalous dispersion depends on: the volume fraction of DEP in the medium, concentration of particles in DEP (i.e., $\alpha$), and the distribution of DEP-size (i.e., the $\Lambda$ distribution). Under a homogeneous particle distribution, the volume fraction of DEP in the medium is equivalent to the parameter $\alpha$. The results from our simulation (also captured by the model) with four different $\alpha$ values show that an increasing $\alpha$ results in an increasingly heavier tail in the BTC, but without any effect on its shape (see Fig. 6a). The distribution of DEP-size relates to the structural heterogeneity of the medium. We examine the effect of $\Lambda$ by considering a Gamma distribution $f(\Lambda) = \Lambda^\kappa e^{-\Lambda/\Lambda_m}$ and varying the exponent $\kappa \in [-1, 0, 1]$. Compared to $\kappa = 0$ (a homogeneous system), a positive exponent (frequent larger DEPs) widens and a negative exponent (frequent smaller DEPs) shrinks the anomalous tail of the the BTC (see Fig. 6b). Further, this model is extendable to a porous medium with pore-size heterogeneity (a wide $\lambda$ distribution) by incorporating it in the distribution of $\Lambda = \lambda^2/\mathcal{A}^2$.

In these scenarios, we further examine the emergence of anomalous dispersion due to particles escaping from the DEPs via a parameter $\mathcal{R} = F_{\text{DEP}}(t)/F(t)$ (Eq. (3)). The effects of $Pe$, $\alpha$, and $\Lambda$ on the evolution of $\mathcal{R}$ are shown in Fig. 6c–e, respectively. In all cases, the anomalous dispersion emerges ($\mathcal{R} \longrightarrow 1$) after the removal of the first pore-volume primarily comprising of particles originating in the TPs. When the Péclet number is large, particles remain trapped in the dead-end vortices for relatively longer times. Hence, besides the longer anomalous tail in the BTC (as observed for $Pe = 6800$ in Fig. 3c), this results in a delay in the onset of anomalous dispersion compared to a smaller $Pe$ (see Fig. 6a). With other parameters fixed, the anomalous transport

emerges earlier for a larger value of $\alpha$ (see Fig. 6c), and nearly at the same time for all three distributions in $\Lambda$ (see Fig. 6d).

A natural porous system may exhibit additional complexities including the presence of ions, $O_2$, salinity gradients in the carrier fluid, the shape and motility of suspended particles (e.g., bacteria and other micro-swimmers). Although such conditions are beyond the scope of this study, the fundamental mechanism presented here will provide a general benchmark for future investigations in the relevant disciplines involving dead-end pores in porous media. For instance, the prolonged vortex-induced trapping of substances in the dead-end pores could create nutrient or salinity rich micro-spots, leading to preferential motion in bacteria via chemotaxis[11] and in colloidal suspensions via diffusiophoresis[57]. Furthermore, the quantified control parameters will help biomedical industries to design new strategies to sustain resident drugs and other desired substances in tissue cavities[58].

In summary, our findings shed new light on a fundamental mechanism governing particle dispersion in disordered porous systems characterized by dead-end pores of fluid stagnation. Classical descriptions overlook dead-end flow structures assuming that fluid stagnation does not play a significant role on macroscopic transport[25,59]. Such diffusion based models cannot quantitatively predict the observed 2/3 power-law decay of the macroscopic BTC. The anomalous arrival time distribution is caused by the observed pore-scale vortical and laminar flow structures within the dead-end pores. Here, in the case of colloidal transport, we have shown a quantitative link connecting the characteristic decay of macroscopic BTC to the delay associated with a hopping and rolling of particles along the streamlines inside the dead-end pores. The particle-diffusivity, initial particle distribution and the size distribution of dead-end pores control the BTC long tail cutoff, but they do not change the observed power-law scaling. Since the model described here is not dependent on the system dimensionality, it can be readily applicable to disordered porous systems encountered in natural soil, and other biological and environmental systems.

## Methods

**Medium characterization**. To obtain maps of the pore-size ($\lambda$) distribution and the segregation index ($\zeta$) of the model porous matrix, we analyzed the image representing the medium geometry (Fig. 1a). This image is a binary two-dimensional array of pixels that distinguishes the pore space (pixel value = 1, white) from the grains (pixel value = 0, black). First, we labeled each grain and identified its boundary. The binary image was then skeletonized to obtain the 1-pixel width representation of the pore space (see Supplementary Fig. 2). For a specific skeleton location, we employed a maximum inscribed circle (MIC) algorithm (see Supplementary Fig. 2) that fits the largest circle on its neighboring grain boundaries based on the Euclidean distance map. By iteratively scanning all skeleton locations with this algorithm, we generated the series of the locally largest circles touching the grain boundaries at three points and spanning across the entire pore space domain. The pore size ($\lambda$) at a specific location was assigned as the average diameter of the overlapping circles around that location. The segregation index ($\zeta$) was assigned at each location as the number of individual, different, grains touched by the inscribed circle at that location. A pixel with $\zeta = 1$ belongs to a dead-end pore and that with $\zeta > 1$ belongs to a transmitting pore (TP). For locations outside the fitted circles the segregation index was obtained via linear 2D interpolation of the neighboring $\zeta$ values.

**Time-lapse video-microscopy**. Time-lapse imaging was performed with an automated inverted microscope (Eclipse Ti2, Nikon) equipped with a CMOS camera (Hamamatsu ORCA flash 4.0, 16-bit). All individual pictures (2048 × 2048 pixels) were recorded at 10X magnification (corresponding to 0.65 $\mu$m/pixel) focusing the microscope optics on the middle horizontal plane of the microfluidic channel. Colloids were imaged by fluorescence microscopy (using a Nikon DAPI filter cube) with an exposure time of 50 ms. Pictures of the flowing colloids were recorded every 5 min close to the outlet by a composite image of 9 individual pictures along the transverse flow direction to cover the entire cross section of the channel outlet.

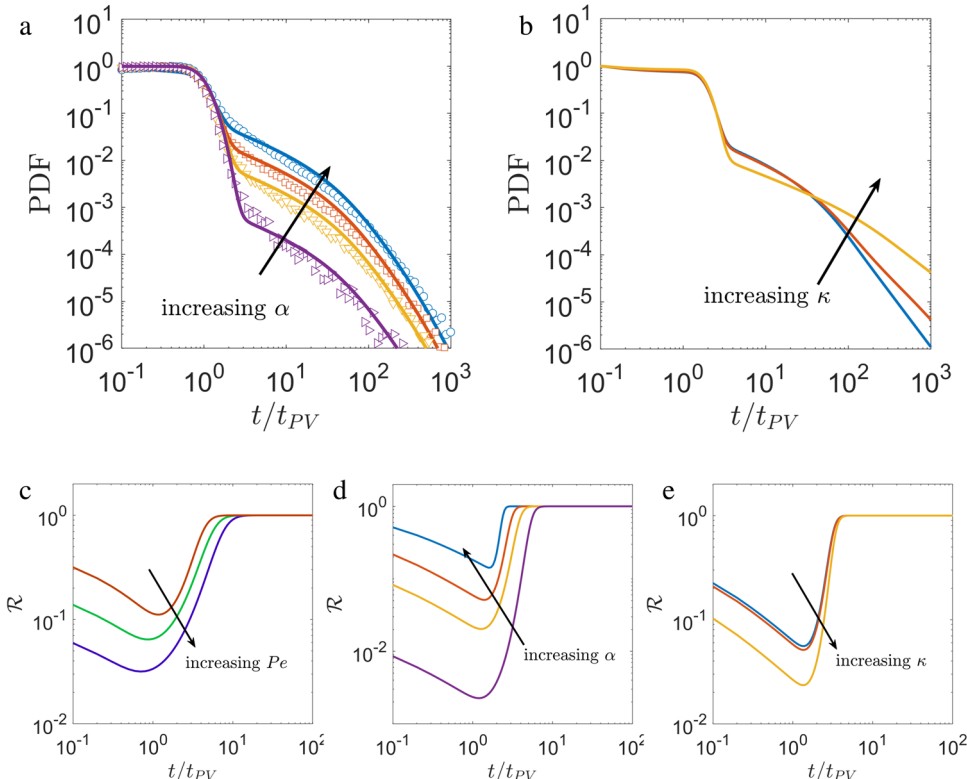

**Fig. 6 Parametric effects on anomalous dispersion of particles through a structurally heterogeneous porous medium. a** Particles escape time probability density function (PDF) based on numerical simulation (symbols) and the analytical model in Eq. (3) (lines) for four values of $\alpha \in$ [0.01, 0.09, 0.22, 0.5] and a Péclet number, $Pe = 680$. **b** Prediction of particle escape time PDF from the analytical model for three $\kappa \in$ [−1, 0, 1] values in the DEP aspect ratios ($\Lambda$) distribution: $\Lambda^{\kappa}e^{-\Lambda/\Lambda_m}$. The onset of anomalous dispersion characterized by $\mathcal{R} = F_{DEP}/F(t)$ as a function of pore-volumes ($t/t_{PV}$) for (**c**). $Pe = 6800$, 680, 68 and $\alpha = 0.22$. **d** $\alpha = 0.01$, 0.09, 0.22, 0.5 and $Pe = 680$. **e** for $\kappa = -1$, 0, 1; $\alpha = 0.22$, $Pe = 680$.

**Measuring BTC**. In order to compute the number $n$ of effluent colloids at the time $t_k$, we first accounted for the deposited particles detected in the field of view by removing the image at time $t_{k+1}$ from the image at time $t_k$, we, then, applied a bandpass filter with a characteristic noise length of 1 pixels to every recorded image in order to smooth the electronic noise associated with camera acquisition. We identified the colloids as individual peaks with a minimum brightness of 5% the pixel depth (16 bit) and minimum diameter of 3 pixels. The BTCs of effluent colloids were obtained as $c = n(t_k)/n_0$, normalizing the measured number of colloids at time $t_k$ by the count $n_0$ at the beginning of the experiment ($t_0$). The physical time $t$, defined as time elapsed since the injection begins, was rescaled by the residence time of one pore volume, defined as $T_{PV} = L\,w\,a\,\phi/Q = 21.8$ min, where $L = 51$ mm is the longitudinal size of the porous system, $w = 7$ mm is channel width, $a = 0.083$ mm is its thickness, $\phi = 0.39$ is the medium porosity and $Q = 0.5$ mm$^3$/min is the imposed flow rate, corresponding to an average (Darcy) velocity of $q = 0.038$ mm/s.

**Classical transport model through homogeneous porous media**. The classical model to describe transport through a relatively homogeneous porous medium is the so-called advection-dispersion framework that expressed mass conservation as[1,2]:

$$\frac{\partial c}{\partial t} = -q\,\frac{\partial c}{\partial x} + D^*\frac{\partial^2 c}{\partial x^2} \qquad (4)$$

where $D^*$ is the macroscopic dispersion coefficient, in mm$^2$/s. Our experimentally measured BTC are well matched by the analytical solution $c(x, t)$ of the above equation (ADE) evaluated at the medium outlet $x = 50$ mm[11] (Fig. 2 2a green dashed line) for times shorter than about 2 $T_{PV}$. For larger times the TP are basically empty and only DEP mass release contribute to the BTC that deviates from exponentially decaying behavior to scale as a heavy-tailed power law.

**Spatial organization of the suspended and deposited colloids**. Profiles of deposited and suspended colloids were computed from the composite images of the entire microfluidic channel at 0, 1, 6, 10, 16, 22, and 44 h. Clusters of connected pixels larger than 3 × 3 pixels were considered as colloids after removing background via an adaptive-thresholding algorithm which chooses a threshold value based on the local mean intensity over an area of 1001 × 1001 pixels (using the Matlab embedded function *adaptthresh*). Colloids detected in consecutive images with overlapping positions have been considered as deposited (Fig. 2d, e green

dots), while the non-overlapping colloids were categorized as suspended (Fig. 2d, e red dots). The local surface occupied by these two classes of colloids has been integrated along transversal slices of 10 μm and normalized by the accounted porous area to measure the suspended and deposited profiles, shown in Fig. 2f. During the experiment, deposited colloids profile are not changing, meaning that the deposition that took place during the saturation process does not affect the transport experiment.

**Numerical simulation to predict colloid transport**. The Stokes flow solution used to model the colloids transport is computed over a domain discretized using a physics based unstructured mesh with approximately $5 \times 10^6$ elements and the element size adapting to the geometry with a minimum of 0.2 μm (see Supplementary Fig. 4). We imposed no-slip boundary conditions at all grain and domain boundaries, a zero reference pressure at the outlet and an uniform flow rate $Q = 0.5$ μL/min at the inlet. The computational resolution is high enough to ensure a divergence free velocity field.

## Data availability

The collected experimental data, simulation results, and the plots data used and discussed in this study are available in the Zenodo database under the https://doi.org/10.5281/zenodo.6617923(https://zenodo.org/record/6617923).

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

## Acknowledgements

P.d.A. acknowledges the support of FET-Open project NARCISO (ID: 828890) and of Swiss National Science Foundation (grant ID 200021 172587). All authors acknowledge the assistance from Faderico Pasotti in performing the PIV experiment, Silvia Guadagnini in measuring the diffusivity of colloids and Monica Bollani for very useful discussions.

## Author contributions

A.D.B., D.S., and P.d.A. designed the research, M.A. and M.B. prepared the de-wetted HPS samples, D.S. performed transport experiments, A.D.B. performed numerical simulations, A.D.B., D.S., and P.d.A. analyzed the data, M.D., A.D.B., D.S., and P.d.A. derived the theoretical model, and all authors wrote the manuscript. A.D.B. and D.S. share the first authorship.

## Competing interests

The authors declare no competing interests.
