## [Peer Review File · Nature Communications]

Structure induced laminar vortices control anomalous dispersion in porous mediaREVIEWER COMMENTS

Reviewer #1 (Remarks to the Author):

This manuscript focuses on exploring the how the particle/solute transport characteristics in dead-end pores in microscopic could impact the transport phenomenon in macroscales. A microfluidic model was first built to conduct experiment, which was composed of both transmitting pores and dead-end pores. The BTC of the solute transport was measured and two distinct regimes were found. The authors then conducted pore-scale numerical investigations using a subset of the experimental domain and found similar results. In order to understand how the flow structures in dead-end pores affect the particle movement, cases for laminar flow over a cavity were simulated and a $-2/3$ power law for the residence time was found for all cases, which is claimed to be caused by the trapped particles in the dead-end pores. Based on these, a theoretical frame was established to result in a stochastic model, so that the total BTC is both controlled by particles in the transmitting pores as well as those in the dead-end pores. This work is original, and the topic is very interesting. The authors did a very comprehensive work (experimental, numerical and theoretical) and tied the observations and conclusions based on them. The conclusion could be of great potential for applications such as subsurface remediation/biological tissues. But I have some concerns/comments regarding the work.

1. The authors used the words "vortices/vortex" throughout the paper and in the title to represent the flow structure in the cavity. Though there is nothing wrong with that, it is somewhat misleading as people always talk about vortices in turbulent flows. While in this work, the authors are focusing on laminar flow only. "Recirculation" might be an alternative word.
2. In the introduction part, the author may want to add the river sediment. The small sediments/fine grains create a closely packed pore network as well. Nutrient flux at the sediment-water interface is a crucial factor affecting nutrient balance and regulating primary productivity in the water. And it is greatly impacted by the trapping and releasing processes related with the sediments.
3. The authors need to work on the section "Numerical simulation" a little bit more. The first few sentence is the motivation to conduct the simulation for a single cavity rather than for the microfluidic system. And why didn't the author use the same collides distribution as measured in the experiment for the simulation but a homogeneous case? Though I noticed it is mentioned in supplemental material that a case with same Alpha has been simulated.
4. Also, for the sentence "Unfortunately, it is very challenging to resolve...such as PIV", the author neglected existed publications on this topic. For example, "Coherent turbulent structures within open-channel lateral cavities" by Mignot and Brevis, who performed PIV measurement for a turbulent flow over a cavity.
5. The main objective of this paper is to understand how the dead-end pores influence the particle residence time in porous media. This is a very interesting topic. But it seems the mechanism of how particles are trapped by the dead-end pores in the first place is important as well, especially considering that the DEPs of 6% to the total volume trapped 22% of the collides in the experiment. In addition, to use the model proposed in the paper, the alpha in equation 3 is a prerequisite. And it changes the tailing significantly as shown in the last figure in SI. However, I didn't find enough discussions/comments on how this parameter could be estimated for other applications.

Reviewer #2 (Remarks to the Author):

The article by Borodloi et al. attempts to quantify the effect of dead-end pores on the dispersion observed in porous media. Overall, the authors perform both experiments and modeling studies, and conclude that the anomalous dispersion can be explained through the dead-end pores. Overall, the results are interesting, and the findings are important. However, in my view, the manuscript needs to be significantly revised before it is suitable for publication. I list several comments below.

1. I found the layout of the manuscript to be very confusing. The authors jump between experiments, numerical simulations, and analytical modeling, which makes it harder to understand

their results. For instance, they present analytical modeling in both Fig. 2 and 3, but they only discuss the analytical modeling in the "theoretical analysis" section. Then in Fig. 4, they introduce a more zoomed in version of the simulation, and then collapse the numerical results with the analytical fit. I had to read the paper several times over to make sense of the different results authors have reported. Frankly, given that the authors did not have any space limitations, more consideration should be given on how to structure the results.

2. The authors also repeatedly show the porous network which only adds to the confusion. Couldn't the authors simply combine the numerous representations of the porous network, experimental and numerical simulations into just one figure?

3. The main results author intend to highlight is that dead-end pores can transport the solute faster because of "structure-induced vortices". If so, I am not sure why authors don't explore the effect further. At least from their theoretical model, authors should be able to teach us the volume fraction of dead-end pores required to make the anomalous dispersion important. From an experimental perspective, could this volume fraction of dead-end pores be linked to tortuosity? The manuscript falls short of explaining these important effects.

4. I also found the analytical derivation in supplementary information to be somewhat arbitrary. Authors choose to approximate stream-function without providing a formal justification. Their main result of the residence time distribution depends on this approximation. In Fig. 7 of SI, the authors give a more nuanced results for the stream function, but I am not sure why the full expression was not averaged systematically.

5. The authors also completely skip over the flow details inside the dead-end pores obtained from their simulations. Given the title of the manuscript is "structure-induced vorticity", authors should quantify vorticity and provide us with more insights into the fluid flow within the dead-end pores. What happens to vorticity within a dead-end pore? How does vorticity change with the aspect ratio?

6. Finally, authors should also provide more implications of their results. In the current form, the manuscript just lists out what they found and how they can collapse their data for BTCs. However, clearly, there is more information from their model in the SI. Given that anomalous dispersion is an important topic in porous media, authors should provide more context about how their results can be utilized in the field. Could this analysis also help us derive effective 1-D model by considering the DEPs as time-dependent sources? Discussion and analysis along these lines would be helpful.

In summary, while the results presented here are interesting, the manuscript hasn't been organized and analysis and discussion lacks the depth required for a publication in Nature Communications.

Reviewer #3 (Remarks to the Author):

Review for manuscript #NCOMMS-21-50637

Title: Structure induced vortices control anomalous dispersion in porous media

Summary:

This paper combines complimentary microfluidic experiments and direct numerical simulations of porous media, using a novel network of transmitting and dead-end pores to visualize colloid particle motions and to measure high resolution breakthrough curves. Two regimes of transport are identified. The first is normal and found to be created by elution of particles transported in transmitting pores. The second is anomalous and produced by the slow release of particles in dead end pores. A theoretical stochastic model is proposed that is capable of reproducing the observed behaviors very accurately.

The work here presented is novel and of maximum relevance to all disciplines dealing with porous media transport. The state-of-the-art experiments produced high quality data, and the theoretical model is at the forefront of current research. The successful combination of the two with a mechanistic understanding of the system's underpinnings makes an excellent contribution that is more than worthy of publication in Nature Communications. This is a very important advance, given how critical it is to properly model the scaling and duration of heavy tails in many systems.

The relatively minor comments below should be addressed to ensure that the story is perfectly clear to the reader.

Major comments:

It is not clear to this reviewer why it is important to distinguish between mechanisms for hopping (at low Pe) and rolling (at $Pe \sim 1$) across streamlines of the vortex for particles to escape from individual DEPs into the backbone of TPs. Both mechanisms result in particles jumping streamlines. If these two specific mechanisms matter as much as has been emphasized, then they may need to be more explicitly accounted for in the model.

The mass transfer between DEPs and TPs is discussed as a unidirectional process (i.e., DEP \rightarrow TP). Why do the authors neglect the possibility that particles may also transfer from TP to DEP within the pore network? How (in)significant is trapping of particles in DEPs downstream from their initial position?

The authors conclude that particle diffusivity (to hop and roll out of vortex streamlines in DEP) and size distribution of DEPs control the long tail cutoff of breakthrough curves, but do not affect the power-law scaling. However, their assessment of Pe on BTC shown in SI. Figure 5 contradicts this statement. Here, Pe is varied by changing D (defined as diffusion coefficient in line 68), which shows that as Pe increases, the diffusion decreases, and the power-law scaling decreases (tails become heavier) because of DEP trapping. Then, their analysis of α , the fraction of particles initially located in DEP vs TPs, shows that the cutoff where the anomalous tailing begins is controlled well by the initial condition. Please clarify what are the local/global controls for cutoff and scaling of the system-wide BTC heavy tail.

Minor comments:

Line 38: Referring to the structural characteristic of dead-end pores as "depth" while that for transmitting-pores as "size" makes it difficult to compare against each other. Moreover, the data on DEP depth in Figure 1 shows the PDF of effective aspect ratio ($\Lambda = A/\lambda_m^2$), which is not a metric of depth at all. Please provide a more consistent structural characteristic to define these two types of pores.

Line 64: The breakthrough circuit to create a sharp injection front is not clearly explained, including the reference of something called "2 a B-2-3-4".

Line 131: Simulations for two- and three-dimensional geometries of a single DEP generalization are provided and compared against microfluidic experiments that are truly three-dimensional. Although the simulations of both dimensionalities are consistent, only those of dimensions matching the experiment (3D) should be used to compare against experimental data.

The material in the SI is not referenced consistently, and at times inaccurately, in the main text. Some sections refer to specific SI figures, while others refer to the SI generically. Please err on the side of being overly explicit of the SI subsection/figure referenced. E.g., line 137 calls on SI Figure 5, but this figure does not show trajectory behavior discussed in the text.

Reviewer #4 (Remarks to the Author):

The article establishes a link between the microscopic transport mechanisms within dead-end

pores and the overall macroscopic colloidal transport in a model complex porous medium using a combination of experimental, analytical, and numerical analyses.

While the elements of the analyses presented here appear technically sound, I believe that the current state in which the study is designed and presented addresses only a very specific problem in the field of fluid mechanics and transport in porous media without considering any of the common physico-chemical complexities and geometrical non-homogeneities in the medium and/or the particles. For this reason (see below for more details), I do not believe that this article fits the scope of the Journal.

- The motivating experimental observation is made in an artificial fully saturated porous medium. This design of the experimental porous media is not simple enough to allow in depth particle tracking analyses, such as those presented in Fig. 4 of the manuscript. At the same time, it is not clear how/why this specific level of complexity is chosen. Would the three-dimensionality impact the observation? Does the ratio of the transmitting to dead-end pore volumes play any role? Is homogeneity important?

- It would help broadening the scope of the article if the initial observation is made in a common natural porous medium, or the link between the microfluidic setup and typical natural environments is clearly discussed. For example, how does the porosity compare with natural systems?

- In figure 2c and 3b, the experimental measurements are always overlaid with the curve presenting the analytical model, that is presented much later in the text. This is confusing, especially in figure 2 as it seems like the tailing trend is simply predictable from the start.

- The tailing in the arrival time is reported here in the absence of salinity and in general charge gradient that are commonly present in transport problems in porous media. In such conditions, previous observations report the opposite behaviour in which more particles are trapped within the dead-end pores due to hopping from transmitting-pores' streamlines onto the recirculating dead-end pores' streamlines. I understand that this is not the focus of the current study but considering that such complexities are ubiquitously present in nature and biology, a discussion on the potential effect of other physico-chemically stimulated transport mechanisms on the overall trends presented here would make the article more accessible to a larger community.

In general, the link to a larger picture and more realistic natural and biological scenarios is largely missing, even though the abstract and the introduction claim to address a common geometry/flow problem in geology, biology, and drug delivery.

REVIEWER 1

This manuscript focuses on exploring the how the particle/solute transport characteristics in dead-end pores in microscopic could impact the transport phenomenon in macroscales. A microfluidic model was first built to conduct experiment, which was composed of both transmitting pores and dead-end pores. The BTC of the solute transport was measured and two distinct regimes were found. The authors then conducted pore-scale numerical investigations using a subset of the experimental domain and found similar results. In order to understand how the flow structures in dead-end pores affect the particle movement, cases for laminar flow over a cavity were simulated and a $-2/3$ power law for the residence time was found for all cases, which is claimed to be caused by the trapped particles in the dead-end pores. Based on these, a theoretical frame was established to result in a stochastic model, so that the total BTC is both controlled by particles in the transmitting pores as well as those in the dead-end pores. This work is original, and the topic is very interesting. The authors did a very comprehensive work (experimental, numerical and theoretical) and tied the observations and conclusions based on them. The conclusion could be of great potential for applications such as subsurface remediation/biological tissues. But I have some concerns/comments regarding the work.

We thank the reviewer for his/her careful reading of the paper and the valuable comments that have helped improve the manuscript. We also thank the reviewer for a globally positive assessment of this work. Below, we address a detailed point by point response to all the remarks that have been raised. We also point out how we revised the paper in accordance with the reviewer's suggestions. The modification are quoted below and are highlighted in the track changes file.

1. The authors used the words “vortices/vortex” throughout the paper and in the title to represent the flow structure in the cavity. Though there is nothing wrong with that, it is somewhat misleading as people always talk about vortices in turbulent flows. While in this work, the authors are focusing on laminar flow only. “Recirculation” might be an alternative word.

We acknowledge the concern raised by the reviewer and agree that the referred two words are widely used to describe turbulent flows. However, we would like to emphasize that “vortices/vortex” is a general fluid mechanics term used to characterize an element of fluid that spins about an axis (also prevalent in a wide range of laminar flows). We like the proposition “recirculation” from the reviewer, but we think it rather describes a phenomenon that the fluid element undergoes and does not quite fit into the title. Because of these reasons, we did not remove the word “vortices” from the title of the manuscript. After careful consideration, we made the following implementations in the revised manuscript. We changed the title to “*Structure induced laminar vortices control anomalous dispersion in porous media*”, which we believe will clarify this confusion. We use the word “re-circulation” suggested by the reviewer to describe the vortex the first place it appears in the manuscript using the following sentence (line 138 of the revised manuscript):

Contrary to this simplified view about such systems, we show the existence of complex vortex flow structures characterized by flow recirculation within the DEPs that, coupled with molecular diffusion, control the macroscopic BTC.

2. In the introduction part, the author may want to add the river sediment. The small sediments/fine grains create a closely packed pore network as well. Nutrient flux at the sediment-water interface is a crucial factor affecting nutrient balance and regulating primary productivity in the water. And it is greatly impacted by the trapping and releasing processes related with the sediments.

We thank the reviewer for this suggestion. We have included the following sentence in the introduction to emphasize the importance of this study in river sediment flows (line 21 in the revised manuscript).

On the other hand, morphological diversity introduced by disordered pore structures ..., nutrient transport through river sediments that create a closely packed pore network [],....

3. *The authors need to work on the section “Numerical simulation” a little bit more. The first few sentences is the motivation to conduct the simulation for a single cavity rather than for the microfluidic system. And why didn’t the author use the same collides distribution as measured in the experiment for the simulation but a homogeneous case? Though I noticed it is mentioned in supplemental material that a case with same Alpha has been simulated.*

Based on this and other comments from the reviewers, we significantly revised our manuscript and most of our earlier figures. We clarified the beginning of the simulation section (revised section heading: Pore-scale simulation) by providing our justification for conducting the simulations in the microfluidic system. The beginning of this section reads as (line 107 in the revised manuscript):

Because the fluid velocities within the DEPs are significantly smaller than the ones within the TPs, a detailed understanding of such velocity field requires a multi-scale description. Although Eulerian velocity measurement techniques such as Particle Image Velocimetry (PIV) has been used to quantify the fluid velocity inside cavities (Mignot, 2020), unfortunately it is challenging to simultaneously resolve all the scales of such flow-fields in high-resolution using such methods (see SI: Fig. 1). Herein, we use COMSOL Multiphysics to numerically solve the two-dimensional steady state incompressible Stokes flow equations in a subsection of the microfluidics geometry (see Methods *f* and SI). The domain of this numerical computation is approximately one-fifth in length (11 mm) and the same in width (7 mm), as that in the experiment.

We also show the effect of three Péclet numbers on the BTC for the value of α that matches the experimental condition (Fig. 3*b*). Further, we include a discussion related to the effect of α in Fig. 6*a* of the manuscript.

4. *Also, for the sentence “Unfortunately, it is very challenging to resolve. . . such as PIV”, the author neglected existed publications on this topic. For example, “Coherent turbulent structures within open-channel lateral cavities” by Mignot and Brevis, who performed PIV measurement for a turbulent flow over a cavity.*

We agree that there has been carefully done PIV studies on turbulent flow through cavities. Thanks to the reviewer, we have included the suggested paper in our reference (line 110). The referred sentence was mainly used to emphasize the challenge related to resolving the dynamic range in velocity in a laminar cavity flow. For example, the velocity in the transmitting pore could be several orders of magnitude larger than the one inside the cavity, making it difficult to simultaneously resolve all the scales of velocity via conventional Eulerian field measurement techniques such as PIV. This is clarified in the Supplementary Fig. 7 *a* and *b*.

5. **The main objective of this paper is to understand how the dead-end pores influence the particle residence time in porous media. This is a very interesting topic. But it seems the mechanism of how particles are trapped by the dead-end pores in the first place is important as well, especially considering that the DEPs of 6% to the total volume trapped 22% of the collides in the experiment. In addition, to use the model proposed in the paper, the alpha in equation 3 is a prerequisite. And it changes the tailing significantly as shown in the last figure in SI. However, I didn’t find enough discussions/comments on how this parameter could be estimated for other applications.**

We thank the reviewer for this comment. Although the topic of particle trapping is slightly out of the scope for this paper, we agree that it is of interest to many geophysical and biological flows. To investigate this, we computed the entry rate of particles into the dead end pores in a clean porous system from a continuous injection of flux weighted particles at the medium entrance. The imposed flow rate is 0.2 $\mu\text{L}/\text{min}$, which is the same condition used to set up the initial condition for the BTC experiment in the paper. We performed this simulation only upto a few pore volumes since it is computationally very expensive. The figure below shows the temporal variation of particle number ratio

α between the DEPs and the TPs. The curve asymptotically increases toward a homogeneous distribution given by $\alpha = 0.09$ by approximately 20 pore volumes (~ 20 hours). We did not include this result in our manuscript, because we believe that this topic requires further investigation. As we discuss in the revised version of the manuscript, even after careful preparation of the suspension by matching at best the suspension-colloids density, during several hours of slow injection, within the small volume of the syringe some colloids sedimentation happens. This reduces the injected concentration of the colloids with time: thus, towards the end of the experiment preparation within the TPs there is a slightly smaller concentration (since it is removed by the injected one) than in the DEPs.

FIG. 1. Trapping of particles in DEP

We also compute the number of particles re-entering a dead-end pore during its trajectory through the porous system. As now discussed in the revised manuscript, we find that it is only a very small fraction of total injected particles (less than 1%). Based on 1 million particle simulations for three different Péclet numbers, the particles re-entry is insignificant. For $Pe = 68, 680$ and 6800 , we found the re-entry of particles are 0.3%, 0.1%, and 0.05%, respectively of the total initial particles (shown in Fig 3c). At line 73 of the revised manuscript, we added:

We quantify the fractional particle number density in DEP ($\alpha = n_{DEP}/n_{total}$) by analyzing the fluorescence signal in the microscopic image of the medium after 24 hours and identifying particles trapped inside the DEPs (see SI). The measured $\alpha = 0.22$ is approximately twice the volume fraction of total DEPs inside the system. This difference is attributed to the fact that, towards the end of the 24 hours some particles in the middle of the injection syringe have settled under gravity causing dilution to the initial concentration of the particles.

And at line 129 of the revised manuscript, we added:

In addition, we consider the possibility of a particle re-entering into a DEP, by counting such instances (see Fig. 3 *c*). The re-entry event is significantly small ($< 0.1\%$) primarily due to a) only a small fraction of particles get close to the DEP entrance and b) the total volume fraction of DEP covers less than 10% of the total porous matrix.

REVIEWER 2

The article by Borodloi et al. attempts to quantify the effect of dead-end pores on the dispersion observed in porous media. Overall, the authors perform both experiments and modeling studies, and conclude that the anomalous dispersion can be explained through the dead-end pores. Overall, the results are interesting, and the findings are important. However, in my view, the manuscript needs to be significantly revised before it is suitable for publication. I list several comments below.

We thank the reviewer for his/her careful reading of the paper and the valuable comments that have helped improve the readability and content of the manuscript. We also thank the reviewer for considering this work interesting and important. Based on these comments, we significantly revised the manuscript. Below, we address a detailed point by point response to all the remarks that have been raised. We also point out how we revised the manuscript in accordance with the reviewer's suggestions. The modifications are quoted below and are highlighted in the track changes file.

1. I found the layout of the manuscript to be very confusing. The authors jump between experiments, numerical simulations, and analytical modeling, which makes it harder to understand their results. For instance, they present analytical modeling in both Fig. 2 and 3, but they only discuss the analytical modeling in the "theoretical analysis" section. Then in Fig. 4, they introduce a more zoomed in version of the simulation, and then collapse the numerical results with the analytical fit. I had to read the paper several times over to make sense of the different results authors have reported. Frankly, given that the authors did not have any space limitations, more consideration should be given on how to structure the results.

We considerably revised the layout of the manuscript to improve its readability. Most importantly, we revised the structure of the results section, modified and rearranged the figures. Instead of having separate sections for experiment, simulation and analytical modeling, we divide the result section into the following concept-oriented subsections. The revised manuscript has two additional figures.

- (a) Structure characterization: we characterize the dual feature (dead-end and transmitting pores) of the adopted porous structure (Fig.1) whose relevance to various natural systems has been previously discussed.
- (b) Macroscopic transport experiment: we describe the microfluidic experiment to measure the macroscopic transport, and discuss the influence of dead-end pores on the breakthrough curve (BTC) anomalous tail (Fig.2).
- (c) Pore scale simulations: to overcome experimental limitations on the resolution of the velocity field across several orders of magnitude, we use pore-scale simulations to characterize the flow field in the system. We compute the BTC for three different Péclet numbers (Fig.3 *a – c*) and discuss the impact of particle diffusivity on the anomalous tail of the BTC.
- (d) Structure induced vortices in pore-scale transport: based on our simulation and experiment, we demonstrate the structure induced vortices inside a dead-end pore. We describe the physical mechanism of a particle hopping across and rolling along the closed streamlines of a vortex to escape from a dead-end pore (Fig.3 *d – f*).
- (e) Bridging from pore-scale feature to macroscopic transport: herein, we explore the flow inside single rectangular 3D cavities as a model for the dead-end pores. We show that the mechanism of particle transport from the cavity via "hopping and rolling" along streamlines leads to a Gamma-distribution in the heavy-tailed breakthrough curve (Fig.4). We analytically justify the 2/3 power-law decay exponent in the Gamma distribution by deriving the residence time distribution of particles in a vortex bounded by two parallel surfaces. Based on the transport study on a single cavity, we build a continuous time random walk (CTRW) model that captures the macroscopic BTC both from simulation and experiment (Fig 5), at all times. Using this model and additional simulations, we explore further the impact of the controlling parameters (new Fig. 6): i) the fraction of particles in DEP vs.

TP (α), ii) the distribution of the DEP aspect ratios (Λ) and iii) the Péclet number.

2. The authors also repeatedly show the porous network which only adds to the confusion. Couldn't the authors simply combine the numerous representations of the porous network, experimental and numerical simulations into just one figure?

We agree that numerous figures showing different porous networks in the system could create confusion. However, instead of combining all representations into one figure, we think that it is more helpful to show the porous network in the respective individual figure for a better visual guidance. To improve clarity, we provide consistency between experiment and simulation, to our best, we show now the same regions that are clearly distinguishable in all figures.

3. The main results author intend to highlight is that dead-end pores can transport the solute faster because of "structure-induced vortices". If so, I am not sure why authors dont explore the effect further. At least from their theoretical model, authors should be able to teach us the volume fraction of dead-end pores required to make the anomalous dispersion important. From an experimental perspective, could this volume fraction of dead-end pores be linked to tortuosity? The manuscript falls short of explaining these important effects.

Regarding the first part of this comment, we would like to first clarify that the overall transport is faster in the current flow system compared to a completely diffusion based system (advection is turned off). This is evident in the steeper slope in the tail of the breakthrough curve (BTC) compared to a diffusion based system (with slope of $-1/2$). However, it is important to note that in presence of flow, the dead-end pore vortices hinder the particle release. In this scenario, a particle is trapped in closed vortex streamlines, and the only way a particle can escape the vortex is to hop across these streamlines via diffusion. We quantify this in terms of Péclet number (Pe) and show individual trajectories for three Pe separated by an order of magnitude.

Overall, the anomalous transport depends on the initial particle distribution between TP and DEP (α), the medium structural properties (Λ distribution controlling the BTC cut off) and the Pe number. Under a homogeneous particle distribution, the volume fraction of DEP in the medium is equivalent to the parameter α . Based on this interesting comment, we carefully quantified the effect of α , Λ distribution and Pe to provide an additional discussion analyzing their effects on anomalous breakthrough curves. We added a new figure (Fig. 6) discussing these effects to the main text. At line 203 of the revised manuscript, we write:

In addition to molecular diffusion of the transported particles and the average flow (i.e. the Péclet number), the observed anomalous dispersion depends on: the volume fraction of DEP in the medium, concentration of particles in DEP (i.e. α), and the distribution of DEP-size (i.e the Λ distribution). Under a homogeneous particle distribution, the volume fraction of DEP in the medium is equivalent to the parameter α ...

We do not think there is any connection between the hydraulic tortuosity of the porous material to the anomalous BTC presented herein. However, since the dead-end pores in our system are not always straight, we conducted an analysis by fixing the effective aspect ratio ($\Lambda = 4$) but changing the shape of the cavity into a tortuous L-shape. We show below (see figure 1) that the tortuosity does not effect the breakthrough curve. This is due to the fact that the power-law scaling in the BTC is primarily controlled by the vortex near the entrance. We added this comment to the revised manuscript (line 180):

Additionally, we examine the effect of the DEP shape by considering a tortuous cavity and find that the tortuosity of a DEP does not affect the BTC. The shape-independent characteristics is attributed to the fact that the power-law scaling in the BTC is primarily controlled by the fastest vortex near the entrance.

In addition, we quantify the emergence of anomalous dispersion in terms of the new parameter \mathcal{R} by taking the relative contribution from the DEPs to the entire BTC, from our analytical model. At line 214 of the revised manuscript, we add:

FIG. 1. **Distribution of particle escape time is independent of tortuosity of the dead-end pore.** Modulus of velocity [mm/s] in log-scale superposed with selected streamlines in a two-dimensional **a**, straight and **b**, tortuous dead-end pore with aspect ratio $\Lambda = 4$ connected to a channel, and **c**, the corresponding probability density function (PDF) of escape time of particles.

In these scenarios, we further examine the emergence of anomalous dispersion due to particles escaping from the DEPs via a parameter $\mathcal{R} = F_{\text{DEP}}/F(t)$, where F_{DEP} is the BTC of particles originated in the DEPs (see, equation 3). The effects of Pe , α , and Λ on the evolution of \mathcal{R} are shown in Figs. 6(c, d, e), respectively. In all cases, the anomalous dispersion emerges ($\mathcal{R} \rightarrow 1$) after the first pore-volume of dispersal of particles originating in the TPs with $\mathcal{R} \ll 1$. When the Péclet number is large, particles remain trapped in the dead-end vortices for relatively longer times. Hence, besides the longer anomalous tail in the BTC (as observed for $Pe=6800$ in Fig. 3c), this results in a delay in the onset of anomalous dispersion compared to a smaller Pe (see Fig. 6a). With other parameters fixed, the anomalous transport emerges earlier for a larger value of α (see Fig. 6c), and nearly at the same time for all three distributions in Λ (see Fig. 6d).

4. *I also found the analytical derivation in supplementary information to be somewhat arbitrary. Authors choose to approximate stream-function without providing a formal justification. Their main result of the residence time distribution depends on this approximation. In Fig. 7 of SI, the authors give a more nuanced results for the stream function, but I am not sure why the full expression was not averaged systematically.*

We assume the reviewer refers to the stream-function approximation by

$$\psi(x) \approx \sigma x^2/2 \quad (1)$$

close to the no-slip boundary. This approximation is unique for laminar flow in the vicinity of a no-slip boundary. The flow field behaves as $\mathbf{v}(\mathbf{x}) = (0, \sigma x)^\top$. As an example consider Poiseuille flow in a channel, whose velocity profile is $v(x) = v_0 x(2 - x/a)/a$ with v_0 the maximum velocity in the channel, and a the half-diameter. Close to the wall, the flow field is $v(x) = \sigma x$ with $\sigma = v_0/a$. The travel time is dominated by the low velocities close to the no-slip boundaries. Therefore, the quadratic approximation of the stream-function captures the dominant behavior, see also the articles by Young et al. (Phys. Fluid, 1989) and Bouchaud and Georges, both of which are cited in the manuscript.

The stream-function shown in supplementary Figure is merely a model stream-function to illustrate the spacing of streamlines.

5. *The authors also completely skip over the flow details inside the dead-end pores obtained from their simulations. Given the title of the manuscript is “structure-induced vorticity”, authors should quantify vorticity and provide us with more insights into the fluid flow within the dead-end pores. What happens to vorticity within a dead-end pore? How does vorticity change with the aspect ratio?*

We would like to clarify here that our analysis refers to the structure induced vortex structure and not vorticity. The vorticity (which relates to the curl of velocity) contains two parts: the one due to fluid shear and the one due to fluid rotation. Hence, vorticity can arise independent of a vortex structure. For example, the laminar flow close to the boundaries will generate strong vorticity due to fluid shear, but it does not contain any vortex structure due to the lack of the rotation element. We computed the fluid vorticity in the dead-end pore and it decays exponentially along its length, as shown in Fig. 2a below.

FIG. 2. **Vorticity field inside a cavity.** **a**, Magnitude of vorticity along the centerline of cavity with aspect ratio $\Lambda = 1, 2, 4, 8$; **b**, Contours of vorticity normalized by U/λ_m , where U is the local velocity magnitude and λ_m is the size of the pore.

A vortex structure is characterized by fluid rotation with a core stagnant region. The quantification of a vortex structure is a debatable topic and an interesting discussion can be found in “Definitions of vortex vector and vortex” by Tian et al. (JFM, 2018). Visually a vortex core can be identified by the peaks in the normalized vorticity ($\omega\lambda_m/U$) plot, (where ω and U are the local vorticity and the local absolute velocity, respectively) as shown in Fig 2 for dead-end pores of different aspect ratios.

We do not think that this discussion about vorticity has any relevance to our results and, therefore, we prefer not to add any additional detail about that in our manuscript.

6. *Finally, authors should also provide more implications of their results. In the current form, the manuscript just lists out what they found and how they can collapse their data for BTCs. However, clearly, there is more information from their model in the SI. Given that anomalous dispersion is an important topic in porous media, authors should provide more context about how their results can be utilized in the field. Could this analysis also help us derive effective 1-D model by considering the DEPs as time-dependent sources? Discussion and analysis along these lines would be helpful.*

We thank the reviewer for this comment. We now provide a schematic in our modified manuscript to illustrate the proposed 1-D model that considers the DEPs as time-dependent source (see Fig 5a in the manuscript). We use this model to further explore the effect of dead-end pore distribution on the BTC. We vary the aspect ratio distribution and show the effect on BTC (see Fig 6b in the manuscript). We show that the higher the frequency of larger Λ values, the longer is the tail. At line 209 of the revised manuscript, we add:

We examine the effect of Λ by considering a Gamma distribution $f(\Lambda) = \Lambda^\kappa e^{-\Lambda/\Lambda_m}$ and varying the exponent $\kappa \in [-1, 0, 1]$. Compared to $\kappa = 0$ (a homogeneous system), a positive exponent (more frequent larger DEPs) widens and a negative exponent (more frequent smaller DEPs) shrinks the anomalous tail of the the BTC (see Fig. 6b). Further, this model is extendable to a porous medium with pore-size heterogeneity (a wide λ distribution) by incorporating it in the distribution of $\Lambda = \lambda^2/\mathcal{A}^2$.

In summary, while the results presented here are interesting, the manuscript hasn't been organized and analysis and discussion lacks the depth required for a publication in Nature Communications.

We believe that the significant manuscript revision we did, following the reviewer suggestions (including new simulations, new figures and extended introduction and discussion), makes this contribution of interest for the broad readership of Nature Communications.

REVIEWER 3

This paper combines complimentary microfluidic experiments and direct numerical simulations of porous media, using a novel network of transmitting and dead-end pores to visualize colloid particle motions and to measure high resolution breakthrough curves. Two regimes of transport are identified. The first is normal and found to be created by elution of particles transported in transmitting pores. The second is anomalous and produced by the slow release of particles in dead end pores. A theoretical stochastic model is proposed that is capable of reproducing the observed behaviors very accurately. The work here presented is novel and of maximum relevance to all disciplines dealing with porous media transport. The state-of-the-art experiments produced high quality data, and the theoretical model is at the forefront of current research. The successful combination of the two with a mechanistic understanding of the system's underpinnings makes an excellent contribution that is more than worthy of publication in Nature Communications. This is a very important advance, given how critical it is to properly model the scaling and duration of heavy tails in many systems. The relatively minor comments below should be addressed to ensure that the story is perfectly clear to the reader.

We sincerely thank the reviewer for his/her careful reading of the paper and for considering this work novel and of relevance to various disciplines. We appreciate the comments provided by the reviewer, which have helped improve the manuscript. Below, we address a detailed point by point response to all the remarks that have been raised. We also point out how we revised the paper in accordance with the reviewer's suggestions. The modification are quoted below and are highlighted in the track changes file.

1. It is not clear to this reviewer why it is important to distinguish between mechanisms for hopping (at low Pe) and rolling (at $Pe \sim 1$) across streamlines of the vortex for particles to escape from individual DEPs into the backbone of TPs. Both mechanisms result in particles jumping streamlines. If these two specific mechanisms matter as much as has been emphasized, then they may need to be more explicitly accounted for in the model.

While describing what happens within a given vortex, we meant by hopping the mechanism by which particles jump from one streamline to another one. Instead, with rolling we identify the mechanism by which particles follow a closed streamline within a vortex. They are, thus, very different mechanisms. Without hopping across the streamlines, a particle is likely to be infinitely trapped in a closed loop vortex. We introduced an additional figure (see Fig. 3f) that shows the trajectories of a particle for three different Péclet numbers and we added at line 150:

This mechanism is demonstrated in Fig. 3f by plotting the trajectory of a particle initiated deep inside a DEP for three Péclet numbers ($Pe = 6800, 680, 68$)...

2. The mass transfer between DEPs and TPs is discussed as a unidirectional process (i.e., DEP/TP). Why do the authors neglect the possibility that particles may also transfer from TP to DEP within the pore network? How (in)significant is trapping of particles in DEPs downstream from their initial position?

We thank the reviewer for this interesting comment. Based on this suggestion, we conducted some extra simulation to track the number of re-entry of particles. Based on 1 million particle simulations for three different Peclet numbers, we find that re-entry of particles is insignificant. For $Pe = 68, 680, 6800$, we found the re-entry of particle are 0.3%, 0.1 %, and 0.05%, respectively of the total initial particles. We provide the following sentences at line 129 of the revised manuscript highlighting this observation.

In addition, we consider the possibility of a particle re-entering into a DEP, by counting such instances (see Fig. 3 c). The re-entry event is significantly small ($< 0.1\%$) primarily due to a) only a small fraction of

particles get close to the DEP entrance and b) the total volume fraction of DEP covers less than 10% of the total porous matrix.

3. *The authors conclude that particle diffusivity (to hop and roll out of vortex streamlines in DEP) and size distribution of DEPs control the long tail cutoff of breakthrough curves, but do not affect the power-law scaling. However, their assessment of Pe on BTC shown in SI. Figure 5 contradicts this statement. Here, Pe is varied by changing D (defined as diffusion coefficient in line 68), which shows that as Pe increases, the diffusion decreases, and the power-law scaling decreases (tails become heavier) because of DEP trapping. Then, their analysis of α , the fraction of particles initially located in DEP vs TPs, shows that the cutoff where the anomalous tailing begins is controlled well by the initial condition. Please clarify what are the local/global controls for cutoff and scaling of the system-wide BTC heavy tail.*

The 2/3 decay scaling that we observed is determined by the coupling between hopping across and rolling along streamlines, as derived by our CTRW physical model. The late-time cut-off is controlled by the diffusion time over the distributed DEP sizes. We now clarify this in the revised version of the manuscript since we also carefully investigated the role of initial conditions (α), medium structural properties (Λ distribution controlling the BTC cut off) and the system Pe number to provide a thorough analysis on their effect on anomalous breakthrough curves. This analysis appears in Fig. 6 and the Discussion section, as follows.

In addition to molecular diffusion of the transported particles and the average flow (i.e. the Péclet number), the observed anomalous dispersion depends on: the volume fraction of DEP in the medium, concentration of particles in DEP (i.e. α), and the distribution of DEP-size (i.e. the Λ distribution). Under a homogeneous particle distribution, the volume fraction of DEP in the medium is equivalent to the parameter α . The results from our simulation (also captured by the model) with four different α values show that an increasing α results in an increasingly heavier tail in the BTC, but without any effect on its shape (see Fig. 6a). The distribution of DEP-size relates to the structural heterogeneity of the medium. We examine the effect of Λ by considering a Gamma distribution $f(\Lambda) = \Lambda^\kappa e^{-\Lambda/\Lambda_m}$ and varying the exponent $\kappa \in [-1, 0, 1]$. Compared to $\kappa = 0$ (a homogeneous system), a positive exponent (more frequent larger DEPs) widens and a negative exponent (more frequent smaller DEPs) shrinks the anomalous tail of the the BTC (see Fig. 6b). Further, this model is extendable to a porous medium with pore-size heterogeneity (a wide λ distribution) by incorporating it in the distribution of $\Lambda = \lambda^2/\mathcal{A}^2$.

Minor comments

1. *Line 38: Referring to the structural characteristic of dead-end pores as “depth” while that for transmitting-pores as “size” makes it difficult to compare against each other. Moreover, the data on DEP depth in Figure 1 shows the PDF of effective aspect ratio ($\Lambda = A/\lambda_m^2$), which is not a metric of depth at all. Please provide a more consistent structural characteristic to define these two types of pores.*

We thank the reviewer for this comment: we now consistently refer throughout the revised manuscript to DEP-size and remove the word depth. The statistical distribution of the DEP effective aspect ratio Λ is a very important metric in this context since it controls the distribution of cut-off for each individual DEP escape time probability density function (a gamma distribution) that must be statistically averaged to get the overall BTC. In the modified manuscript, we discuss these effects in a new figure (Fig. 6b) and add a paragraph at the beginning of the Discussion section.

2. *Line 64: The breakthrough circuit to create a sharp injection front is not clearly explained, including the reference of something called “2 a B-2-3-4”.*

We clarified this at line 79 of the revised manuscript, as follows.

Next, we withdraw a clean solution with the same density (Solution B: 1:1 milliQ water and D₂O mixture) but without the colloidal suspension through the inlet via a cleaning circuit (see Fig. 2 a B-2-1-4): the medium outlet stays closed while we withdraw liquid from a hole next to the inlet, as discussed in [11]. This generates a particle-free sharp front at the inlet without perturbing the particles within the porous matrix.

3. *Line 131: Simulations for two- and three-dimensional geometries of a single DEP generalization are provided and compared against microfluidic experiments that are truly three-dimensional. Although the simulations of both dimensionalities are consistent, only those of dimensions matching the experiment (3D) should be used to compare against experimental data.*

We agree with the reviewer: we use numerical simulations on single DEP to identify the fundamental mechanisms of hopping across and rolling along closed vortex streamlines in a single cavity. We performed 3D simulations of single cavities of different aspect ratio, but, due to technical limitations, we could not perform a 3D simulation through the whole medium (which we believe is not necessary). Thus, We do not compare experiments to simulations, but rather our model with experiments and our model to our simulations, showing that dimensionality does not matter in this context but rather the fundamental mechanisms of rolling and hopping that take place in 2D as well as in 3D.

4. *The material in the SI is not referenced consistently, and at times inaccurately, in the main text. Some sections refer to specific SI figures, while others refer to the SI generically. Please err on the side of being overly explicit of the SI subsection/figure referenced. E.g., line 137 calls on SI Figure 5, but this figure does not show trajectory behavior discussed in the text.*

We revised the manuscript accordingly.

REVIEWER 4

The article establishes a link between the microscopic transport mechanisms within dead-end pores and the overall macroscopic colloidal transport in a model complex porous medium using a combination of experimental, analytical, and numerical analyses. While the elements of the analyses presented here appear technically sound, I believe that the current state in which the study is designed and presented addresses only a very specific problem in the field of fluid mechanics and transport in porous media without considering any of the common physico-chemical complexities and geometrical non-homogeneities in the medium and/or the particles. For this reason (see below for more details), I do not believe that this article fits the scope of the Journal.

We thank the reviewer for his/her constructive criticisms that have helped improve the manuscript. It is not in the scope of this article to cover all the physico-chemical complexities taking place in a natural porous system. The geometrical non-homogeneity of the medium manifested as dead-end pores remains the primary focus of this paper. We have extensively modified the manuscript, adding further analysis on possible variations of the system from the experimental conditions and discussed their implications in detail. Below, we address a detailed point by point response to all the remarks that have been raised. We point out how we revised the paper in accordance with the reviewer's suggestions. The modification are quoted below and are highlighted in the track changes file.

1. The motivating experimental observation is made in an artificial fully saturated porous medium. This design of the experimental porous media is not simple enough to allow in depth particle tracking analyses, such as those presented in Fig. 4 of the manuscript. At the same time, it is not clear how/why this specific level of complexity is chosen. Would the three-dimensionality impact the observation? Does the ratio of the transmitting to dead-end pore volumes play any role? Is homogeneity important? Can our system be linked to the natural system?

One of our findings is that the dimensionality of the medium is not important for the observed anomalous dispersion of particles from dead-end pores. To clarify this, we refer to the following sentences of the revised manuscript.

At line 169:

The trajectories for both 3D and 2D cavities (see also SI: Fig. 5) show behavior similar to the ones in the DEP of the porous medium: the particle initially diffuses isotropically ($Pe^* \ll 1$) until it reaches the upper part of the cavity, where its motion is a result of the competition between advection along the vortex streamlines and diffusion that promotes streamline exchange ($Pe^* \sim 1$). Finally, the particle exits the cavity and follows the channel (TP) flow with $Pe^* \gg 1$.

At line 179:

The BTCs obtained for 2D cavities also exhibit an identical collapse (see Supplementary Fig. 6). This suggests that the thickness of the cavity does not affect the transport of particles from a dead-end pore.

We believe that these observations are of importance, given that a natural porous system is also not simple enough to perform 3-D particle tracking experiment. We choose this system based on its disordered hyperuniform character that is prevalent in many natural systems. We emphasize this in the introduction adding the following sentences from line 33 of the revised manuscript.

As an archetype of such structural heterogeneity, we exploit spinodal-like morphology, that emerge from first-order phase transitions [1] in binary systems featuring a miscibility gap and undergoing phase separation, as described by Cahn-Hilliard dynamics originally developed for metallic alloys [2–4]. Such morphology of connected structures are the common ground for a plethora of natural and artificial systems undergoing

spontaneous pattern formation, including volcanic rocks [5, 6], semiconductors [7–9], metallic thin films [10], metal hydrides [11–15], liquid-gas [16], polymers [17–20], proteins [21] and tumor cells [22, 23].

In reference to the ratio of the transmitting to dead-end pore volumes, this effect is parameterized as the ratio (α) of particles initially trapped in the DEPs vs. in the TPs. The question related to homogeneity is addressed by the variation in the distribution of DEPs. Both these effects are examined in Fig 6. We add a related discussion in the beginning of Discussion section in the manuscript. We refer to the paragraph starting with line 203:

In addition to molecular diffusion of the transported particles and the average flow (i.e. the Péclet number), the observed anomalous dispersion depends on: the volume fraction of DEP in the medium, concentration of particles in DEP (i.e. α), and the distribution of DEP-size (i.e the Λ distribution)...

2. It would help broadening the scope of the article if the initial observation is made in a common natural porous medium, or the link between the microfluidic setup and typical natural environments is clearly discussed. For example, how does the porosity compare with natural systems?

The purpose of this research work is to investigate the fundamental mechanisms underlying the anomalous transport in a disordered porous structure comprising dead-end pores. Such structures are ubiquitously present in soil [24], gut tissue [25], and polymeric filters [26], to name a few. We chose an archetype model that is representative of many natural porous systems (as referred to in our previous response). Various parameters such as porosity ($\phi = 0.39$) and volume fraction of dead-end pores ($< 10\%$) of the chosen model are carefully matched with natural soil [27]. The mechanisms discovered in this work has been overlooked in earlier studies, and we believe that this work will provide a benchmark to extend into the respective areas of science with additional complexities.

3. In figure 2c and 3b, the experimental measurements are always overlaid with the curve presenting the analytical model, that is presented much later in the text. This is confusing, especially in figure 2 as it seems like the tailing trend is simply predictable from the start.

We removed the overlay of the analytical model from Fig. 2c and Fig. 3b. In the modified manuscript, the model prediction appears in Fig. 5b, Fig.6a and Supplementary Fig. 5. We believe that this eliminates the confusion.

4. The tailing in the arrival time is reported here in the absence of salinity and in general charge gradient that are commonly present in transport problems in porous media. In such conditions, previous observations report the opposite behaviour in which more particles are trapped within the dead-end pores due to hopping from transmitting-pores' streamlines onto the recirculating dead-end pores' streamlines. I understand that this is not the focus of the current study but considering that such complexities are ubiquitously present in nature and biology, a discussion on the potential effect of other physico-chemically stimulated transport mechanisms on the overall trends presented here would make the article more accessible to a larger community.

As the reviewer rightly pointed out, it is not the focus of our paper to examine the effects of salinity and charge gradients in the system. Based on this comment, we added the following in the Discussion section, at line 222.

A natural porous system may exhibit additional complexities including the presence of ions, O_2 , salinity gradients in the carrier fluid, the shape and motility of suspended particles (e.g. bacteria and other microswimmers). Although such conditions are beyond the scope of this study, the fundamental mechanism presented here will provide a general benchmark for future investigations in the relevant disciplines involving dead-end pores in porous media. For instance, the prolonged vortex-induced trapping of substances in the dead-end pores could create nutrient or salinity rich micro-spots, leading to preferential motion in bacteria via chemotaxis [28] and in colloidal suspensions via diffusiophoresis [29]. Furthermore, the quantified control parameters will help biomedical industries to design new strategies to sustain resident drugs and other desired substances in tissue cavities [30].

5. *In general, the link to a larger picture and more realistic natural and biological scenarios is largely missing, even though the abstract and the introduction claim to address a common geometry/flow problem in geology, biology, and drug delivery.*

We addressed this reviewer’s concerns by significantly revising the introduction, adding new results and providing perspectives related to potential applications in the discussion section. As emphasized in these revisions, we believe that the findings from this work are applicable to a wide range of scientific disciplines and of interest for the broad readership of Nature Communications.

-
- [1] Kurt Binder. Theory of first-order phase transitions. *Reports on progress in physics*, 50(7):783, 1987.
- [2] John W Cahn and John E Hilliard. Free energy of a nonuniform system. i. interfacial free energy. *The Journal of chemical physics*, 28(2):258–267, 1958.
- [3] John W Cahn and John E Hilliard. Free energy of a nonuniform system. iii. nucleation in a two-component incompressible fluid. *The Journal of chemical physics*, 31(3):688–699, 1959.
- [4] John W Cahn. On spinodal decomposition in cubic crystals. *Acta metallurgica*, 10(3):179–183, 1962.
- [5] Piers PK Smith. Spinodal decomposition in a titanomagnetite. *American Mineralogist*, 65(9-10):1038–1043, 1980.
- [6] Michele Menna, Mario Tribaudino, and Alberto Renzulli. Al-si order and spinodal decomposition texture of a sanidine from igneous clasts of stromboli (southern italy): insights into the timing between the emplacement of a shallow basic sheet intrusion and the eruption of related ejecta. *European Journal of Mineralogy*, 20(2):183–190, 2008.
- [7] I-hsiu Ho and GB Stringfellow. Solid phase immiscibility in gainn. *Applied Physics Letters*, 69(18):2701–2703, 1996.
- [8] Aaron M Holder, Sebastian Siol, Paul F Ndione, Haowei Peng, Ann M Deml, Bethany E Matthews, Laura T Schelhas, Michael F Toney, Roy G Gordon, William Tumas, et al. Novel phase diagram behavior and materials design in heterostructural semiconductor alloys. *Science advances*, 3(6):e1700270, 2017.
- [9] M. Salvalaglio, M. Bouabdellaoui, M. Bollani, A. Benali, L. Favre, J.B. Claude, J. Wenger, P. de Anna, F. Intonti, A. Voigt, and M. Abbarchi. Hyperuniform monocrystalline structures by spinodal solid-state dewetting. *Phys. Rev. Lett.*, 125:126101, 2020.
- [10] Stephan Herminghaus, Karin Jacobs, Klaus Mecke, Jorg Bischof, Andreas Fery, Mohammed Ibn-Elhaj, and Stefan Schlagowski. Spinodal dewetting in liquid crystal and liquid metal films. *Science*, 282(5390):916–919, 1998.
- [11] Shiva Rudraraju, Anton Van der Ven, and Krishna Garikipati. Mechanochemical spinodal decomposition: a phenomenological theory of phase transformations in multi-component, crystalline solids. *npj Computational Materials*, 2(1):1–9, 2016.
- [12] Ashutosh Sharma and Rajesh Khanna. Pattern formation in unstable thin liquid films. *Physical Review Letters*, 81(16):3463, 1998.
- [13] Richard V Craster and Omar K Matar. Dynamics and stability of thin liquid films. *Reviews of modern physics*, 81(3):1131, 2009.
- [14] Adam F Wallace, Lester O Hedges, Alejandro Fernandez-Martinez, Paolo Raiteri, Julian D Gale, Glenn A Waychunas, Stephen Whitelam, Jillian F Banfield, and James J De Yoreo. Microscopic evidence for liquid-liquid separation in super-saturated caco3 solutions. *Science*, 341(6148):885–889, 2013.
- [15] Ken-ichiro Murata and Hajime Tanaka. Impact of surface roughness on liquid-liquid transition. *Science advances*, 3(2):e1602209, 2017.
- [16] Vincent Testard, Ludovic Berthier, and Walter Kob. Influence of the glass transition on the liquid-gas spinodal decomposition. *Physical review letters*, 106(12):125702, 2011.
- [17] R Xie, Alamgir Karim, Jack F Douglas, Charles C Han, and Robert A Weiss. Spinodal dewetting of thin polymer films. *Physical Review Letters*, 81(6):1251, 1998.
- [18] Benjamin P Lee, Jack F Douglas, and Sharon C Glotzer. Filler-induced composition waves in phase-separating polymer blends. *Physical Review E*, 60(5):5812, 1999.
- [19] Hiroaki Sai, Kwan Wee Tan, Kahyun Hur, Emily Asenath-Smith, Robert Hovden, Yi Jiang, Mark Riccio, David A Muller, Veit Elser, Lara A Estroff, et al. Hierarchical porous polymer scaffolds from block copolymers. *Science*, 341(6145):530–534, 2013.
- [20] Michael Schmitt, Jianan Zhang, Jaejun Lee, Bongjoon Lee, Xin Ning, Ren Zhang, Alamgir Karim, Robert F Davis,

- Krzysztof Matyjaszewski, and Michael R Bockstaller. Polymer ligand-induced autonomous sorting and reversible phase separation in binary particle blends. *Science advances*, 2(12):e1601484, 2016.
- [21] Frederic Cardinaux, Thomas Gibaud, Anna Stradner, and Peter Schurtenberger. Interplay between spinodal decomposition and glass formation in proteins exhibiting short-range attractions. *Physical Review Letters*, 99(11):118301, 2007.
- [22] C Chatelain, T Balois, Pasquale Ciarletta, and M Ben Amar. Emergence of microstructural patterns in skin cancer: a phase separation analysis in a binary mixture. *New Journal of Physics*, 13(11):115013, 2011.
- [23] Abramo Agosti, Paola Francesca Antonietti, Pasquale Ciarletta, Maurizio Grasselli, and Marco Verani. A cahn-hilliard-type equation with application to tumor growth dynamics. *Mathematical Methods in the Applied Sciences*, 40(18):7598–7626, 2017.
- [24] Amandine Erktan, Dani Or, and Stefan Scheu. The physical structure of soil: Determinant and consequence of trophic interactions. *Soil Biology and Biochemistry*, 148:107876, 2020.
- [25] Siegfried Hapfelmeier, Melissa A. E. Lawson, Emma Slack, Jorum K. Kirundi, Maaïke Stoel, Mathias Heikenwalder, Julia Cahenzli, Yuliya Velykoredko, Maria L. Balmer, Kathrin Endt, Markus B. Geuking, Roy Curtiss, Kathy D. McCoy, and Andrew J. Macpherson. Reversible microbial colonization of germ-free mice reveals the dynamics of iga immune responses. *Science*, 328(5986):1705–1709, 2010.
- [26] William A. Phillip, Rachel Mika Dorin, Jörg Werner, Eric M. V. Hoek, Ulrich Wiesner, and Menachem Elimelech. Tuning structure and properties of graded triblock terpolymer-based mesoporous and hybrid films. *Nano Letters*, 11(7):2892–2900, 2011. PMID: 21648394.
- [27] N. Nishiyama, T. Yokoyama, and S. Takeuchi. Size distributions of pore water and entrapped air during drying-infiltration processes of sandstone characterized by water-expulsion porosimetry. *Water resour. Res.*, 48:W09556, 2012.
- [28] de Anna Pietro, Pahlavan Amir A., Yawata Yutaka, Stocker Roman, and Juanes Ruben. Chemotaxis under flow disorder shapes microbial dispersion in porous media. *Nat. Phys.*, 17:68–73, 2021.
- [29] Sangwoo Shin, Eujin Um, Benedikt Sabass, Jesse T. Ault, Mohammad Rahimi, Patrick B. Warren, and Howard A. Stone. Size-dependent control of colloid transport via solute gradients in dead-end channels. *Proceedings of the National Academy of Sciences*, 113(2):257–261, 2016.
- [30] Libero Italo Giannola, Flavia Maria Sutera, and Viviana De Caro. Physical methods to promote drug delivery on mucosal tissues of the oral cavity. *Expert Opinion on Drug Delivery*, 10(10):1449–1462, 2013. PMID: 23802558.

REVIEWERS' COMMENTS

Reviewer #1 (Remarks to the Author):

All my comments have been detailed addressed. Thus, I support this manuscript to be published in Nature Communications.

Reviewer #2 (Remarks to the Author):

I thank the authors for taking my comments seriously and for significantly revising the manuscript. Overall, the restructuring of the manuscript has certainly made the article more appealing and easier to follow. I also appreciate that authors expanded their results to explore more variations. Having said that, I noticed a few things:

1. Fig. 1 caption is wrong; the authors probably forgot to update the caption.
2. Fig. 1 caption: probably best to use either μm for pore sizes given the caption
3. Fig. 2 caption: Is "b" really a single dead-end pore, or more a zoomed-in view?
4. Fig. 2 (d), (e) – If I understand it correctly, the authors are measuring concentration of colloids relative to initial values. Therefore, perhaps better to clarify that this is concentration somewhere... "Dispersed colloids" and "mobile colloids" by itself is meaningless.
5. Fig.3b - It is probably better to show an arrow stating "Pe" instead of "D".
6. Fig.3 caption – a comma is missing after "e"
7. Fig.3f colorbar could be clarified with the actual values instead of $\log...10^{(-4)}$, $10^{(-3)}...$

Reviewer #4 (Remarks to the Author):

I thank the authors for their effort to revise the manuscript and answer to my previous comments. The manuscript is now significantly restructured to start with experimental observations/measurements, logically transiting to the simulations and finalising with parametric studies. Revisions considered in Fig. 2 of the manuscript and including the new figures (Figs. 5 and 6) help better understand the scope of the work and its potential for broader applications. I am happy to support this work for publication in Nature Communications. I have some minor comments on the revised manuscript listed below:

Please review your manuscript for references to figures. See examples below:

- In the revised manuscript, lines 71-78: please check and update references to figure 1 in your revised manuscript, especially fig. 1e and Fig. 1f.
- In the revised manuscript lines 98-99: please check references to Figs. 2a-b.

Reviewer #2 (Remarks to the Author):

I thank the authors for taking my comments seriously and for significantly revising the manuscript. Overall, the restructuring of the manuscript has certainly made the article more appealing and easier to follow. I also appreciate that authors expanded their results to explore more variations.

We appreciate the support of this reviewer and we are grateful for the time she/he spent reviewing our manuscript and providing inspiring comments and remarks.

Having said that, I noticed a few things:

1. Fig. 1 caption is wrong; the authors probably forgot to update the caption.

The reviewer is correct, we modified the figure caption as: ***Characterization of the model porous structure reveals dual feature of the complex medium.*** ***a***, Binary image of the disordered hyperuniform porous structure: it exhibits a complex pore network (white) interspersed among disordered grains (black) within the system. ***b***, The narrow Probability Density Function (PDF) of the pore size, exhibiting a strong peak about the average value $\lambda_m = 27 \mu\text{m}$. ***c***, A portion of the pore size map (λ , [μm]) highlighting the inscribed circles (cyan) along the porous network that estimate the local pore-size (see Methods a). ***d***, The dual feature of the medium characterized by the transmitting-pores (TP: green) surrounded by multiple grains and the dead-end pores (DEP: magenta) surrounded by a single grain. ***e***, Red dots represent the measured PDF of width to depth aspect ratio for DEPs, defined as the ratios between each DEP area (\mathcal{A}) and the mean pore-space area (λ_m^2); the solid black line is its best fit with a Gamma distribution f_Λ .

2. Fig. 1 caption: probably best to use either um for pore sizes given the caption

We modified as suggested, see comment above.

3. Fig. 2 caption: Is “b” really a single dead-end pore, or more a zoomed-in view?
4. Fig. 2 (d), (e) – If I understand it correctly, the authors are measuring concentration of colloids relative to initial values. Therefore, perhaps better to clarify that this is concentration somewhere. . . “Dispersed colloids” and “mobile colloids” by itself is meaningless.

The reviewer is correct, we modified this caption and the figure itself as: **Dual geometric feature of the medium leads to two distinct regimes in the breakthrough curve of colloidal particles.** **a**, Schematic of the experimental setup and **b**, three-dimensional representation of the colloidal suspension within the porous structure. **c**, Experimental breakthrough curve (BTC, blue dots) computed as the $c(t)/c_0$, where c_0 is the injected colloidal numerical density, and $c(t)$ is the measured density eluted at time t . The dashed line represents the analytical solution of advection-dispersion equation (eq. 26 in Supplementary Information III). The dotted line represents the prediction from diffusion based scaling (without advection). **d**, Two snapshots of suspended (red) and deposited (green) colloids just before and 6 hour after injection, respectively; TPs are represented as green areas while DEPs as magenta. **e**, Profile of deposited (above) and suspended particles (below) along the channel length acquired at three different times (0, 1 and 6 hrs) after the start of the experiment.

5. Fig.3b - It is probably better to show an arrow stating "Pe" instead of "D".
6. Fig.3 caption – a comma is missing after "e"
7. Fig.3f colorbar could be clarified with the actual values instead of $\log \dots 10^{(-4)}, 10^{(-3)} \dots$

We agree, thus we modified figure 3 and its caption as suggested. **Computation of velocity field and the role of structure induced vortices on particle dispersal:** **a**, Modulus of the Stokes flow solutions (mm/s in log-scale) in a subsection ($1/5^{\text{th}}$ in length) of the porous medium used in the experiment is superposed to particles that initially occupy the TP (green) and DEP (magenta) (enlarged view in the inset); **b**, Probability density function (PDF) of particle escape time (equivalent to the BTC) versus normalized time (t/t_{PV}) obtained from particle tracking in the simulated velocity field with $\alpha = 0.22$ for three Péclet numbers ($Pe = 68, 680, 6800$). The magenta and green shades distinguish the regions of the BTC for $Pe = 680$ contributed by the particles shown in corresponding colors in the inset of **a**. The long tail of the PDF is contributed by particles originating in the DEPs; **c**, Fraction of particle number density re-entering a dead-end pore for the three Péclet numbers; **d**, Qualitative identification of vortex structure inside a DEP captured by time-stacking experimental images taken at $Pe \sim 10^5$; **e**, Close view of the vortex structures inside a DEP from the simulated velocity field; **f**, An individual trajectory of a particle originated at the magenta dot and leaving the DEP for $Pe = 6800, 680$ and 68 . The trajectories are color-coded with a local Péclet number $Pe^* = \lambda_m v_p / D_m$ in log-scale.

Reviewer #4 (Remarks to the Author):

I thank the authors for their effort to revise the manuscript and answer to my previous comments. The manuscript is now significantly restructured to start with experimental observations/measurements, logically transiting to the simulations and finalising with parametric studies. Revisions considered in Fig. 2 of the manuscript and including the new figures (Figs. 5 and 6) help better understand the scope of the work and its potential for broader applications. I am happy to support this work for publication in Nature Communications.

We appreciate the support of this reviewer and we are grateful for the time she/he spent reviewing our manuscript and providing inspiring comments and remarks.

I have some minor comments on the revised manuscript listed below:

Please review your manuscript for references to figures. See examples below: - In the revised manuscript, lines 71-78: please check and update references to figure 1 in your revised manuscript,

especially fig. 1e and Fig. 1f. - In the revised manuscript lines 98-99: please check references to Figs. 2a-b.

We corrected the typos reported and we checked the references to each figure.

Sincerely,

Pietro de Anna and co-authors

Figure 1: NEW FIGURE 2

Figure 2: NEW FIGURE 3